# Uniform as Glass: Gliding over the Pareto Front with Neural Adaptive Preferences

## Abstract

Multiobjective optimization (MOO) is prevalent in numerous real-world applications, in which a Pareto front (PF) is constructed to display optima under various preferences. Previous methods commonly utilize the set of Pareto objectives (particles) to represent the entire Pareto front. However, the corresponding discrete distribution of the points on the PF is less studied, which may impede the generation of diverse and representative Pareto objectives in previous methods. To bridge the gap, we highlight in this paper the benefits of uniformly distributed Pareto objectives on the PF, which alleviate the limited diversity found in previous multiobjective optimization (MOO) approaches. In particular, we introduce new techniques for measuring and analyzing the uniformity of Pareto objectives, and accordingly propose a new method to generate asymptotically uniform Pareto objectives in an adaptive manner. Our proposed method is validated through experiments on real-world and synthetic problems, which demonstrates its efficacy in generating high-quality uniform Pareto objectives on the Pareto front.

## 1 Introduction

Real-world applications such as recommendation systems (Zheng & Wang, 2022; Jannach, 2022), autonomous agent planning (Xu et al., 2020; Hayes et al., 2022), and industrial design (Schulz et al., 2017; Wang et al., 2011) often involve multiobjective optimization (MOO) problems. For instance, one may expect a robot to strike a balance between its forward speed and energy consumption (Basaklar et al., 2022; Xu et al., 2020). In MOO problems, despite the small

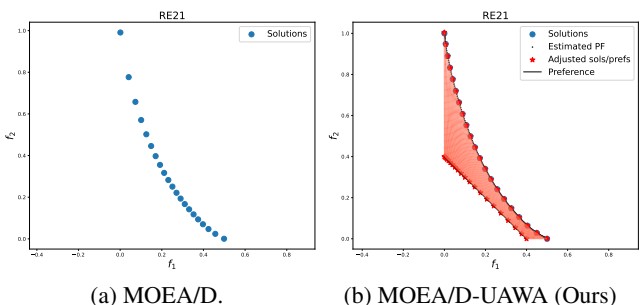

Figure 1: The proposed method generates uniform objectives on the PF. (a) MOEA/D solutions. (b) Proposed solutions.

objective size, achieving optimal values for all objectives is often extremely challenging. Hence, learning the set of Pareto solutions (Miettinen, 1999) that are not dominated by other solutions and provide different trade-offs among objectives, is a preferable choice for addressing MOO problems. An illustrative example of Pareto solutions for a two-objective problem can be seen in Figure 1. The collection of Pareto solutions is referred to as the Pareto set (PS), with their corresponding objective vectors forming the Pareto front (PF).

In the past few decades, a large amount of MOO algorithms have been proposed for constructing a finite set of solutions (dubbed a "population" in MOO) to approximate the Pareto front. The multi-objective evolutionary algorithms (MOEAs) are the most popular methods among them due to their ability to avoid bad local optima and to obtain a set of solutions in a single run (Blank & Deb, 2020; Caramia et al., 2020). The MOEAs can be mainly divided into three different types, namely, the decomposition-based method (e.g., MOEA/D (Zhang & Li, 2007; Zhang et al., 2008; Qi et al., 2014; Ma et al., 2017)), the domination-based method (e.g., NSGAs (Deb et al., 2002a; Ibrahim et al., 2016; Deb & Jain, 2013; Jain & Deb, 2013)), and the indicator-based method (e.g., SMS-EMOA (Beume et al., 2007)).

One crucial challenge for the current MOO research is how to efficiently generate a set of Pareto objectives uniformly distributed on the whole Pareto front. Such a uniform objective set can well represent diverse optimal trade-offs among different objectives, which supports flexible decision-making and is desirable for many real-world applications. Although some efforts have been made, the current MOEAs still struggle to obtain a set of uniformly distributed solutions, especially for practical problems with unbalanced objectives and complicated objective landscapes. An illustration example can be found in Figure 1(a), where the solutions generated by classical MOEA/D are biased to a local region and cannot sufficiently cover the Pareto front.

In this work, we introduce a theoretical tool to measure the uniformity of Pareto objectives on the Pareto front. The results (Proposition 1 and Theorem 2) illustrate the benefits of uniform Pareto objectives for MOO and explain why previous MOO methods are unable to produce uniform Pareto objectives, both theoretically (Theorem 1) and empirically. Based on these findings, we then propose MOEA/D-UAWA, a practical algorithm in the MOEA/D framework to generate Uniform Pareto objectives on the Pareto front utilizing Adaptive Weight Adjustment. The weight adjustment is guided by a neural model on an estimated Pareto front as illustrated in Figure 1(b). Through extensive experiments, we demonstrate that MOEA/D-UAWA consistently produces high-quality and uniform Pareto objectives for both synthetic and real-world MOO problems and achieves better uniformity of Pareto objectives compared to other MOEAs.

The contribution of this paper can be summarized as follows:

1. We present a comprehensive analysis of the uniform Pareto objectives on the Pareto front along with the associated benefits. Additionally, We introduce new tools to measure uniformity and thereby achieve uniform Pareto objectives.
2. We explore the reason behind the inability of previous methods to generate uniformly distributed Pareto objectives. Accordingly, we propose a novel preference adjustment method that utilizes a neural model to represent the Pareto objective distribution, enabling the generation of uniformly distributed solutions on the Pareto front.
3. We perform extensive experiments on synthetic and real-world MOO problems. These experiments demonstrate that our method efficiently generates uniformly distributed Pareto objectives. Compared to previous MOEAs, our method outperforms them in terms of diversity and runtime.

For clarity, all the notations used in this paper can be found in Table 4 in Appendix A.2.

## 2 PRELIMINARIES

In this section, we give a brief description of important concepts in MOO. A multiobjective problem, which optimizes $m$ conflicting objectives, is formally denoted as

$$\min_{x \in \mathscr{X} \subset \mathbb{R}^n} f(x) = (f_1(x), \ldots, f_m(x)), \tag{1}$$

which admits multiple solutions under different preferences on $f_i$'s. For an MOO problem, it is difficult to compare two solutions simply. The concepts of domination and Pareto solutions are thereby introduced.

**Definition 1** (Domination). *A solution $x^{(a)}$ **dominates** $x^{(b)}$ if there exists $i \in [m]$ such that $f_i(x^{(a)}) < f_i(x^{(b)})$ and $\forall j \in [m] \setminus \{i\}, f_j(x^{(a)}) \leq f_j(x^{(b)})$.*

**Definition 2** (Pareto solution). *A solution $x$ is a Pareto solution if no other solution $x' \in \mathscr{X}$ dominates it. The set of all Pareto solutions is denoted as the Pareto set PS, and its image set $\mathcal{T}$, where $\mathcal{T} = (f \circ \text{PS})$ is referred to as the Pareto front (PF).*

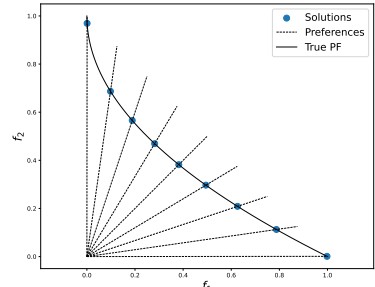

Figure 2: An $\lambda$-exact Pareto solution is the intersection between the Pareto front and vector $\lambda$.

The dominance of a solution $x^{(a)}$ over another solution $x^{(b)}$ implies that $x^{(a)}$ is strictly superior to $x^{(b)}$, which indicates that $x^{(b)}$ cannot be regarded as an optimum in MOO. We additionally explain the concept of weakly Pareto solution, which will be used later in this paper: $x^{(a)}$ is a weakly Pareto solution if there is no other solution $x^{(b)} \in \mathscr{X}$ such that $f(x^{(b)}) \prec f(x^{(a)})$, where $f(x^{(b)}) \prec f(x^{(a)})$ means $f_i(x^{(b)}) < f_i(x^{(a)})$ for all $i \in [m]$.

**Definition 3** (Pareto objective). *For a Pareto solution $x$, $y = f(x) \in \mathbb{R}^m$ is the Pareto objective on the Pareto front $\mathcal{T}$.*

It is very difficult to directly optimize all $m$ objective $f(x)$ due to their conflicting nature. A more practical approach is to convert the objective vector $f(x)$ into a single objective subproblem via the aggregation function $g(\cdot, \lambda)$ with a specific preference $\lambda$. This function takes a preference vector $\lambda \in \Omega$ as arguments to generate a specific Pareto solution (see the rigid definition in Appendix A.1), where $\Omega$ is the support set of preference vectors. Many scalarization methods have been proposed in the past few decades, and this work focuses on the the modified Tchebycheff ("mtche" in short) aggregation function (Ma et al., 2017) with attractive property:

$$g^{\text{mtche}}(y, \lambda) = \max_{i \in [m]} \left\{ \frac{y_i - z_i}{\lambda_i} \right\}, \quad g^{\text{mtche}}(\cdot, \lambda) : \mathbb{R}^m \mapsto \mathbb{R}, \quad (2)$$

to produces $\lambda$-*exact Pareto solutions* (Mahapatra & Rajan, 2020) under mild conditions, where $z$ is a reference point. The optimal Pareto objective $y^*$ for the above problem (2) follows the pattern $\frac{y_1^* - z_1}{\lambda_1} = \ldots = \frac{y_m^* - z_m}{\lambda_m} = C$ with some positive constant $C$ (as shown in Figure 2).

## 3 RELATED WORK

Since we use an adaptive preference adjustment (AWA) by a neural model, we shall discuss the difference between previous AWAs in Section 3.1. Then we show how previous MOO methods use a neural model and discuss the differences with them in Section 3.2. Finally, we discuss the benefits of the proposed method compared with gradient-based MOO methods in Section 3.3.

### 3.1 MOEA/D WITH ADAPTIVE PREFERENCE ADJUSTMENT (MOEA/D-AWA)

MOO researchers have proposed to use adaptive weight adjustments to achieve a more diverse Pareto objectives, including PaLam (Siwei et al., 2011), Adaptive-MOEA/D (Li & Landa-Silva, 2011), and MOEA/D-AWA (Qi et al., 2014). Compared to our proposed method, previous methods mainly use manually-crafted rules to adjust preferences, e.g., MOEA/D-AWA removes the most crowded solution and retains the most sparse solution. The proposed method is instead motivated by a theoretical analysis of the final solution distribution (c.f. Proposition 2), and utilizes a neural model to accelerate optimization process.

### 3.2 LEARNING THE PARETO FRONT/SET BY A NEURAL MODEL

Pareto set learning (PSL) (Navon et al., 2020; Lin et al., 2022) aims to learn the entire Pareto set through a single neural model $x_\beta(\cdot) : \Delta_{m-1} \mapsto \mathbb{R}^n$ [1], which maps a preference vector to a Pareto solution. A PSL model $x_\beta(\cdot)$ is trained by the following loss minimization:

$$\min_\beta \texttt{psl\_loss}(\beta) = \mathbb{E}_{\lambda \sim \text{Unif}(\Delta_{m-1})} \left[ g(f \circ x_\beta(\lambda), \lambda) \right], \quad (3)$$

which is usually optimized by gradient descent (e.g., in Lin et al. (2022)). The gradient involved is computed by the chain rule: $\nabla_\beta \texttt{psl\_loss}(\beta) = \mathbb{E}_{\lambda \sim \text{Unif}(\Delta_{m-1})} \frac{\partial g}{\partial f} \frac{\partial f}{\partial x} \frac{\partial x}{\partial \beta}$. Previous PSL methods sometimes fail to return globally optimal model when the objective $f(\cdot)$ has too many local optima.

We note our proposed Pareto front learning (PFL) model differs from PSL model. Firstly, as mentioned, PSL is a purely gradient-based method that cannot handle local optima, whereas PFL is only adapted as a tool for preference adjustment in evolutionary optimization, which can produce globally optimal solutions in MOO (Zhou et al., 2019). Furthermore, the model size in PSL can be large when the there is a huge amount of decision variables ($n$), whereas the size of PFL remains as PFL addresses the constant objective space. It is suggested by Hamada et al. (2020) and Roy et al. (2023), even in cases where the objectives $f_i$'s and Pareto front are simple and convex, the Pareto set can exhibit a complex structure, posing challenges for PSL.

---

[1] $\Delta_{m-1}$ denotes the $(m\text{-}1)$-dimensional simplex, defined as $\Delta_{m-1} = \{y | \sum_{i=1}^m y_i = 1, \ y_i \geq 0.\}$

## 3.3 GRADIENT-BASED MOO

In the growing trend of MOO, gradient-based methods are mainly adopted to optimize MOO problems. MOO-SVGD (Liu et al., 2021) employs Stein Variational Gradient Descent to achieve a diverse Pareto solution set. Another approach by Chen & Kwok (2022) utilizes a PSL model to optimize diversity or hypervolume of a Pareto solution set. EPO (Mahapatra & Rajan, 2020) and OPT-in-Pareto (Ye & Liu, 2022) focus on finding a single Pareto solution that satisfies specific user requirements. Despite the wide usage, gradient-based MOO methods, as just mentioned in the last subsection, often struggle to produce globally optimal solutions (see a concrete example in Appendix B.6), whereas the proposed method aims to achieve global optimal MOO solutions.

## 4 THE PROPOSED MOEA/D-UAWA METHOD

In this section, we present a novel method for generating uniformly distributed Pareto objectives on the PF. We highlight the theoretical benefit of uniform objectives as Property 1 in Section 4.2 and use Theorem 2 to bound the predicting error of the whole PF by uniform objectives in Section 4.5. Before introducing our method, we provide a theoretical analysis (Section 4.1) on why MOEA/D fails to achieve uniform objectives, and then we develop practical algorithms and discuss the theoretical guarantees of our proposed method in the remaining sections.

## 4.1 DISTRIBUTION OF PARETO OBJECTIVES BY MOEA/D

Previous methods (Deb et al., 2019; Blank et al., 2020) focusing on generating well-spaced (uniform) preferences. We argue that, uniform preferences may lead to non-uniform Pareto objectives. An illustrated example is given in Figure 1. To describe that, we use Theorem 1 to describe the distribution of Pareto objectives by MOEA/D.

We use the preference-to-objective function, denoted as $h(\cdot) : \Delta_{m-1} \mapsto \mathbb{R}^m$, to represent the mapping from a preference $\lambda : \lambda \in \Omega$ [2] to Pareto objectives,

$$y = h(\lambda) = \arg\min_{y \in \mathscr{Y}} g^{\text{mtche}}(y, \lambda), \tag{4}$$

where $\mathscr{Y}$ is the objective space, $\mathscr{Y} = f \circ \mathscr{X}$. We use $\Lambda_N : \Lambda_N = \{\lambda^{(1)}, \ldots, \lambda^{(N)}\}$ to denote a uniform preference set, where $\Lambda_N$ solves the optimization problem, $\max_{\Lambda_N \subset \Omega} \min_{i,j \in [N], i \neq j} \rho(\lambda^{(i)}, \lambda^{(j)})$ (Blank et al., 2020). Here, $\rho(\cdot, \cdot)$ denotes the Euclidean distance between two vectors. We first give the condition of such function $h$ is well defined, i.e., the objective $y$ solves Problem (4) is unique as Lemma 1. This Lemma will also be used to build the PFL model in Section 4.3.

**Lemma 1** (The condition of function $h(\cdot)$ is well defined). *If the objective function $f(\cdot)$ does not have weakly Pareto solutions, the optimal objective $y^*$ that solves Equation 4 is a unique Pareto objective. This implies that the function $h(\cdot)$ is well-defined.*

As function $h$ is well-defined, we provide the following theorem to give the distribution of Pareto objectives $\mathcal{Y}_N : \mathcal{Y}_N = h \circ \Lambda_N$.

**Theorem 1** (Distribution of Pareto objectives). $\widetilde{\mathcal{Y}}_N \xrightarrow{d} h \circ Unif(\Omega)$, where $\widetilde{\mathcal{Y}}_N$ is the uniform distribution over $\mathcal{Y}_N$, $\mathcal{Y}_N = \{y^{(1)}, \ldots, y^{(N)}\}$. The notation "$\xrightarrow{d}$" indicates convergence in distribution, and $Unif(\Omega)$ denotes the uniform distribution over set $\Omega$.

**Remark 1.** Theorem 1 indicates that, only in special cases (e.g., the function $h$ is an affine mapping), uniformity in preference space induces uniformity in the Pareto objective space. We can give a concrete example, by setting $\Omega = \Delta_2$ and the objective function $f$ as the DTLZ1 (Deb et al., 2002b) function. The detailed discussions and proofs for this example are in Appendix C.4. However, even when $h$ is a simple quadratic function, $h \circ Unif(\Omega)$ is not a uniform distribution. The proof of Lemma 1 and Theorem 1 is provided in Appendix C.5 and C.6.

---

[2]$\Omega$ is assumed to be a compact and connected set.

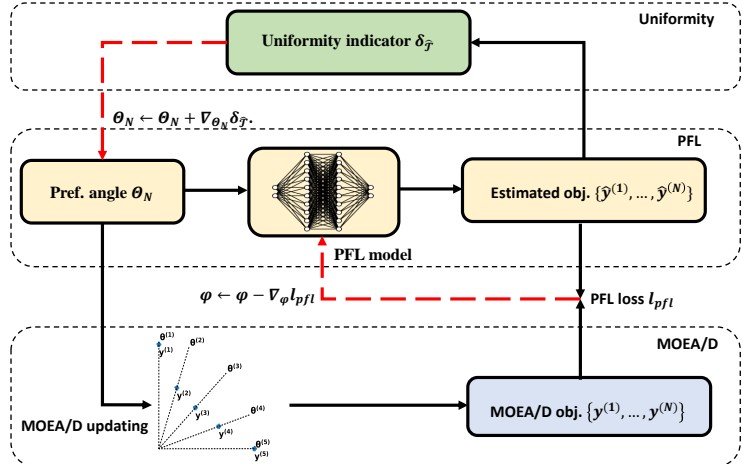

Figure 3: The proposed framework consists of three parts: training the PFL model, updating new preferences (Equation (7)), and MOEA/D. The black solid arrows represent value transfer, and the red dash arrows represent gradient updates.

## 4.2 THE PROPOSED MOEA/D-UAWA FRAMEWORK

**Maximal-Manifold-Separation problem**. To generate uniformly distributed Pareto objectives, we propose to construct the objective configuration $\mathcal{Y}_N^*$ by solving the Maximal-Manifold-Separation (MMS) problem on the Pareto front $\mathcal{T}$,

$$\text{MMS}(\mathcal{T}) = \max_{\mathcal{Y}_N} \delta_{\mathcal{T}} = \max_{\mathcal{Y}_N} \left( \min_{y^{(i)} \neq y^{(j)} \in \mathcal{Y}_N, \mathcal{Y}_N \subset \mathcal{T}} \rho(y^{(i)}, y^{(j)}) \right), \tag{5}$$

where $\rho(\cdot, \cdot)$ denotes the Euclidean distance between two vectors. Intuitively, Problem (5) maximizes the minimum pair-wise distances on $\mathcal{T}$, resulting in a diverse configuration $\mathcal{Y}_N^*$. The optimal separation solving Problem (5) is denoted as $\delta_{\mathcal{T}}^*$.

We formally summarize attractive properties of the optimal configuration $\mathcal{Y}_N^*$ solving Problem equation (5), and invite readers to refer to Appendix C.3 for the proof and additional details. Specifically, Proposition 1 depicts the non-asymptotic property for a fixed sample size $N$, while Proposition 2 yields the asymptotic result.

**Proposition 1** (Covering and $\delta$-Dominance). *Under certain assumptions (Appendix C.3), The optimal configuration $\mathcal{Y}_N^*$ serves as a $\delta_{\mathcal{T}}^*$-packing and a $\delta_{\mathcal{T}}^*$ covering, ensuring that all Pareto objectives in $\mathcal{Y}_N^*$ $\delta_{\mathcal{T}}^*$-dominate any Pareto objectives on the Pareto front $\mathcal{T}$. (The definitions of packing, covering, and $\delta$-dominance can be found in Appendix A.1.) In other words, for every $y' \in (f \circ \Omega)$, there exists $y \in \mathcal{Y}_N^*$ such that $y$ $\delta_{\mathcal{T}}^*$-dominates $y'$.*

**Proposition 2** (Asymptotic Uniformity. (Borodachov et al., 2007)). *$\mathcal{Y}_N^*$ is asymptotically uniform on $\mathcal{T}$, i.e., $\widetilde{\mathcal{Y}}_N^* \xrightarrow{d} Unif(\mathcal{T})$, where $\widetilde{\mathcal{Y}}_N^*$ is the empirical distribution over $\mathcal{Y}_N^*$.*

**Remark 2.** Proposition 1 suggests that the objectives $\mathcal{Y}_N^*$ obtained from solving Problem (5) possess a strong representation ability of the entire Pareto front $\mathcal{T}$. This means that for any objective in $\mathcal{T}$, there exists at least one objective in $\mathcal{Y}_N^*$ that can approximate it within an error tolerance $\delta_{\mathcal{T}}^*$. Additionally, Proposition 2 indicates that as the sample size $n$ increases, $\mathcal{Y}_N^*$ becomes increasingly similar to a uniform distribution.

**Overview of the framework: solving MMS on the unknown Pareto front $\mathcal{T}$.** Since the true PF $\mathcal{T}$ is unknown, as shown in Figure 3, we iteratively estimate $\mathcal{T}$ by Pareto front learning (PFL) and re-pick the preferences for PFL by solving MMS. There are multiple components in the framework. ⓪ The proposed framework is built upon the decomposition-based multiobjective evolutionary algorithm (MOEA/D), where we utilize a set of preference angles $\Theta_N = \{\theta^{(1)}, \dots, \theta^{(N)}\} \subset [0, \frac{\pi}{2}]^{m-1}$ as the inputs for MOEA/D[3]. ① The Pareto Front Learning (PFL) module is then trained using the

---
[3]The angle representation is chosen for its simplicity in optimization with box constraints, and a preference angle along with its corresponding preference is presented in Appendix A.3.

true objectives obtained from the output of MOEA/D. ② Subsequently, the preference angles are updated by optimizing the uniformity indicator.

More detailed descriptions of ① the PFL model and ② the preference update components are provided separately in the subsequent sections. The practical algorithm is implemented as Algorithm 1 and 2 in Appendix A.4, where we also present time complexity analysis. Practically, training time of the PFL and preference adjustment is less than $1s$, which is neglectable compared with MOEAs.

### 4.3 PARETO FRONT LEARNING (PFL)

Table 1: Comparison of PSL and the proposed PFL.

| Method | Mapping function ($n \gg m$) |
|---|---|
| Pareto Set Learning (PSL) (Navon et al., 2020; Chen & Kwok, 2022) | $\Delta_{m-1} \mapsto \mathbb{R}^n$ |
| (Proposed) Pareto front Learning (PFL) | $[0, \frac{\pi}{2}]^{m-1} \mapsto \mathbb{R}^{m-1}$ |

The PFL model, denoted as $h_\phi(\cdot) : [0, \frac{\pi}{2}]^m \mapsto \mathbb{R}^{m-1}$, serves as an approximation of the "preference to objective" function $h(\cdot)$ introduced in Section 4.1. The $h_\phi(\cdot)$ is trained by minimizing the Mean Square Error (MSE) loss between the true Pareto objectives $y$ obtained from MOEA/D and the estimated objectives $h_\phi(\theta)$. For a well-defined training process, two different objectives $y$ and $y'$ cannot both be the optimal value of $g^{\mathrm{mtche}}(\cdot, \lambda)$ for a given preference at the same time. The condition for this property is provided in Lemma 1.

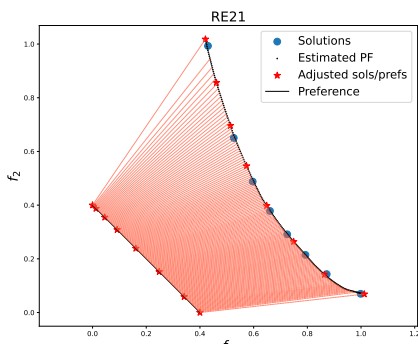

RE21

Figure 4: The original MOEA/D solutions (blue dots) is non-uniform, while solutions solving Problem (5) are uniform (red stars). Uniform Pareto objectives correspond to non-uniform preferences.

We emphasize the necessity of introducing PFL rather than simply applying previous Pareto set learning methods (Equation (3)) (Navon et al., 2020; Lin et al., 2022; Chen & Kwok, 2022). The reasons are twofold. ① PSL simply uses gradient-based methods to optimize the PSL objective function defined in Equation (3). The induced locally optimal solutions make PSL fail on most of ZDT, DTLZ problems. ② The number of decision variables $n$ of an MOO problem can be arbitrarily large, and therefore so can be the size of PSL model. In contrast, the PFL model is constrained in the function space $[0, \frac{\pi}{2}]^{m-1} \mapsto \mathbb{R}^{m-1}$, which implies its complexity is independent of $n$.

We focus on two quantities for a PFL model: (1) the **training loss**, denoted as $l_{\mathrm{pfl}}$, and (2) the **generalization error** for a new angle $\theta'$ after applying the uniform configuration specified in Equation (5). By Allen-Zhu et al. (2019), we can prove the training loss $l_{\mathrm{pfl}}$ converges to a globally optimal solution in polynomial time w.r.t to number of solutions $N$ and width of the neural model. The discussion on the generalization error is deferred to Section 4.5.

### 4.4 PREFERENCE ADJUSTMENT WITH A PFL MODEL

Exactly solving Problem (5) for the optimal configuration is generally an intractable problem (Borodachov et al., 2019). We consider the following surrogate problem in which the preference-to-objective map $h(\cdot)$ is approximated by a neural network $h_\phi(\cdot)$,

$$\mathrm{MMS}(\hat{\mathcal{T}}) = \max_{\Theta_N} \delta_{\hat{\mathcal{T}}} = \max_{\Theta_N} \left( \min_{i \neq j, i, j \in [N]} \rho(h_\phi(\theta^{(i)}), h_\phi(\theta^{(j)})) \right) \tag{6}$$

We find that, given a fixed neural network $h_\phi(\cdot)$ (the PFL model), the preference angles $\theta_i$'s in Problem (6) (as well as the estimated solutions $\hat{y}^{(i)}$'s) can be optimized efficiently via projected gradient ascent method, $\theta^{(i)} \leftarrow \mathrm{Proj}(\theta^{(i)} + \eta \nabla_{\theta^{(i)}} \hat{\delta}_{\mathcal{T}}), i \in [N]$. The $\mathrm{Proj}$ operator projects a preference angle back to its domain $[0, \frac{\pi}{2}]$. The updated rule can be written compactly as the following

equation,

$$\Theta_N \leftarrow \text{Proj}(\Theta_N + \eta \nabla_{\Theta_N} \hat{\delta}_{\mathcal{T}}). \tag{7}$$

Figure 4 demonstrates that, using a PFL model, with only a few Pareto objectives we can effectively estimate the whole Pareto front. The blue dots represent the original Pareto objectives optimized by MOEA/D, which are not uniformly distributed due to Theorem 1; the adjusted preferences are indicated by red stars in the 1-D simplex. After updating preferences using gradient ascent, Pareto objectives are distributed uniformly in the estimated Pareto front, as described in Proposition 2.

## 4.5 PFL GENERALIZATION BOUND

We focus our attention on bounding the generalization error of our proposed methods, namely $\tilde{\epsilon} = |R(\tilde{h}) - \hat{R}(\tilde{h})|$ for an arbitrary $\tilde{h}(\cdot)$. As we have highlighted in Appendix C.1, the population risk of $\hat{h}$ can be controlled via bounding such generalization error. Specifically, we show that the regret error $\tilde{\epsilon} = |R(h) - \hat{R}(h)|$ heavily depends on the margin $\delta_v$. The $\delta_v$ represents for the maximal diameter of the Voronoi cells (Okabe et al., 2009) formed by the Pareto objective $\mathcal{Y}_N = \{y^{(1)}, \ldots, y^{(N)}\}$, where $\mathcal{Y}_N$ solves Equation (5). For the formal definition of Voronoi cells and diameter of a set, please refer to Definitions 8 and 9 in the Appendix A.1.

The complete results are stated as follows (the proof of Theorem 2 is provided in Appendix C.2):

**Theorem 2** (Generalization bound of a PFL model). *We first make some regularity assumptions:*

1. *(Function smoothness). Both $(\tilde{h} - h_*)(\cdot)$ and $h_*^{-1}(\cdot)$ are L- and L'-Lipschitz, respectively, i.e.,*

$$\|(\tilde{h} - h_*)(x_1) - (\tilde{h} - h_*)(x_2)\| \leq L\|x_1 - x_2\|, \quad \forall x_1, x_2 \in \left[0, \frac{\pi}{2}\right]^{m-1},$$
$$\|h_*^{-1}(y_1) - h_*^{-1}(y_2)\| \leq L'\|y_1 - y_2\|, \quad \forall y_1, y_2 \in \mathbb{R}^m, \tag{8}$$

   *where $h_*$ denotes the true mapping function from preferences to objectives.*

2. *(Function upper bound). $\left\|\tilde{h} - h_*\right\|_\infty \leq A$, $\left\|h_*^{-1}\right\|_\infty \leq A'$.*

3. *(Manifold property). We assume $\mathcal{T}$ is a differentiable, compact (m-1)-D manifold, a common assumption found, for example, in (Hillermeier, 2001). Furthermore, we consider $\mathcal{T}$ to be connected, a widely applicable assumption in scenarios such as ZDT 1, 2, 4.*

*For the risk $\tilde{\epsilon} = |R(\tilde{h}) - \hat{R}(\tilde{h})|$, we then have*

$$\tilde{\epsilon} \leq 2\mathcal{H}_{m-1}(\mathcal{T})AA'LL'\delta_v + 2CA^2\sqrt{\mathcal{W}_1(\mathcal{U}, \widetilde{\mathcal{Y}}_N) + \delta_v}, \tag{9}$$

*where $\mathcal{U}$ is the uniform distribution over $\mathcal{T}$, $\widetilde{\mathcal{Y}}_N$ is the empirical distribution of $\mathcal{Y}_N$, $\mathcal{W}_1(\cdot, \cdot)$ is the Wasserstein distance with the $l_1$ norm, $\mathcal{H}_{m-1}(\cdot)$ is the Hausdorff dimension function, and $C$ is some universal constant representing the smoothness of $\mathcal{T}$ (Chae & Walker, 2020, Threom 1).*

**Remark 3.** In Theorem 2 the error bound for $\tilde{\epsilon}$ involves two quantities, the diameter of the Voronoi cell $\delta_v$ and $\mathcal{W}_1(\mathcal{U}, \widetilde{\mathcal{Y}}_N)$. The margin $\delta_v$ is controlled through maximizing the minimal separation distance $\delta_{\mathcal{T}}$. The decaying rate of $\mathcal{W}_1(\mathcal{U}, \widetilde{\mathcal{Y}}_N)$ is impacted by not only the margin $\delta_v$, but also the manifold properties of the Pareto front. I.e., the overall generalization error rate is not completely decided by the margin $\delta_v$. However, by Proposition 2, we still have $\mathcal{W}_1(\mathcal{U}, \widetilde{\mathcal{Y}}_N) \to 0$ since $\widetilde{\mathcal{Y}}_N$ weakly converges to $\mathcal{U}$, and minimizing the margin $\delta_v$ is thus critical to the control of the generalization error $\tilde{\epsilon}$.

## 5 EXPERIMENTS

### 5.1 EXPERIMENT SETTINGS

We validate the effectiveness of the proposed method on various problems, including ZDT1, 2, 4 (Deb et al., 2006), DTLZ 1-2 (Deb et al., 2002b), and real-world testing problems (Tanabe & Ishibuchi, 2020). For the ease of presentation, we normalize the PF of RE37 to $[0, 1]^3$. To test the

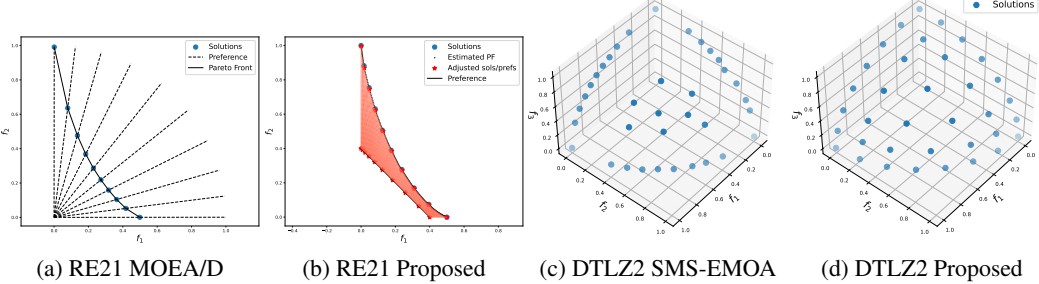

| (a) RE21 MOEA/D | (b) RE21 Proposed | (c) DTLZ2 SMS-EMOA | (d) DTLZ2 Proposed |

Figure 5: Results on RE21 and DTLZ2.

ability of dealing objectives of different scales, we normalize the PFs of RE21 and RE22 to $[0, 0.5] \times [0, 1]$. Problem ZDT4 and DTLZ 1-2 possess numerous locally optima that cannot be identified by gradient-based MOO methods (see Appendix B.6). REX problems are real-world problems with unknown Pareto fronts, demonstrating the capability to handle complex Pareto front shapes. Lastly, there are multiple preference vectors that do not intersect with the RE37 Pareto front, leading to duplicate Pareto objectives by the original MOEA/D. Results on RE37 validate the proposed method can automatically avoid selecting such preferences leading to duplicate solutions, thereby enhancing solution diversity.

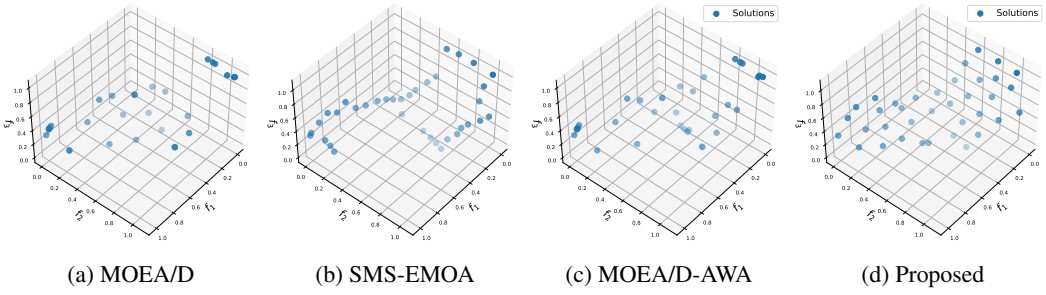

| (a) MOEA/D | (b) SMS-EMOA | (c) MOEA/D-AWA | (d) Proposed |

Figure 6: Results on RE37.

The implementation in this study relies on the pymoo (Blank & Deb, 2020) and PyTorch (Paszke et al., 2019) libraries. We utilize the simulated binary crossover (SBX) operator and polynomial mutation technique (Deb & Beyer, 2001) for MOEA/D-based methods. Following the setting in pymoo, we do not maintain an external population (EP), since it can be computationally and storage intensive, particularly when dealing with many objectives (Li & Landa-Silva, 2011).

Table 2: The running time (every 1k generations) of SMS-EMOA and the proposed method.

| Running Time (m) | DTLZ2 | RE37 |
|---|---|---|
| SMS-EMOA | 1.21 | 3.23 |
| Proposed | 0.56 | 0.71 |

We compare our method with ① the vanilla **MOEA/D** (Zhang & Li, 2007), ② **NSGA2** (Deb et al., 2002a), ③ MOEA/D with adaptive weight adjustment ( **MOEA/D-AWA** ) (Qi et al., 2014), ④ **PaLam** (Siwei et al., 2011), and ⑤ **SMS-EMOA** (Beume et al., 2007). Detailed descriptions of these methods and implementations are deferred to Appendix B.1. To assess the performances, we utilize the `hypervolume` ( `HV` ) ($\uparrow$) (Guerreiro et al., 2020), the `sparsity` ($\downarrow$) (Xu et al., 2020), the `spacing` ($\downarrow$) (Schott, 1995) , the minimal distance on the Pareto front ($\delta_{\mathcal{T}}$) ($\uparrow$), and its soft version ($\tilde{\delta}_{\mathcal{T}}$) ($\uparrow$) indicators. Please refer to Appendix B.2 for detailed descriptions of these indicators.

## 5.2 RESULTS

The average results on five random seeds for the problems are displayed in Table 5 in Appendix B.4, along with the visualization results in Figure 10-17. Detailed discussions on these results are pro-

Table 3: Results on all problems averaged on five random seeds, with the optimal indicators among all problems highlighted in bold. For results on all problems, please refer to Appendix B.4

| Problem | Method | HV ($\uparrow$) | Spacing ($\downarrow$) | Sparsity ($\downarrow$) | $\delta_{\mathcal{T}}$ ($\uparrow$) | $\tilde{\delta}_{\mathcal{T}}$ ($\uparrow$) |
|---------|--------|------|---------|----------|-------|-------|
| RE21 | NSGA2 | 1.226 | 0.066 | 0.022 | 0.024 | -0.063 |
|  | SMS-EMOA | 1.252 | 0.028 | 0.017 | 0.095 | -0.028 |
|  | MOEA/D | 1.246 | 0.086 | 0.024 | 0.072 | -0.058 |
|  | MOEA/D-AWA | 1.250 | 0.028 | 0.017 | 0.088 | -0.031 |
|  | PaLam | **1.252** | 0.025 | 0.017 | 0.101 | -0.027 |
|  | Proposed | 1.252 | **0.002** | **0.016** | **0.123** | **-0.020** |
| RE37 | NSGA2 | 1.051 | 0.069 | 0.005 | 0.013 | -0.140 |
|  | SMS-EMOA | **1.114** | **0.041** | 0.005 | 0.029 | -0.128 |
|  | MOEA/D | 1.052 | 0.072 | 0.013 | 0 | -0.204 |
|  | MOEA/D-AWA | 1.091 | 0.078 | 0.009 | 0.001 | -0.177 |
|  | PaLam | 1.112 | 0.076 | 0.006 | 0 | -0.174 |
|  | Proposed | 1.110 | 0.045 | **0.005** | **0.040** | **-0.086** |

vided. Due to the presence of numerous local optima, gradient-based multi-objective optimization (MOO) methods fail in certain problems. We provide a concrete example of the failure of gradient MOO in Appendix B.6. Key findings from the experiments are summarized in the following section.

① The proposed method achieves the optimal spacing indicator, as shown in Figure 5 and Table 3, for two-objective problems. The spacing indicator is very close to zero, indicating that the distances between adjacent solutions are nearly equal. In comparison to the MOEA/D (Figure 5-(a)), the proposed method generates more uniform objectives. The MOEA/D solutions are denser in the bottom-right area and sparser in the upper-left region, which cannot effectively cover the entire PF when compared to the proposed method.

② We observed that the HV-based method, SMS-EMOA, generated more diverse solutions compared to MOEA/D in RE21 and RE22 (see Table 5 in Appendix B.4). On the RE37 problem, numerous preferences did not intersect with the Pareto front, leading to the production of numerous duplicate Pareto objectives ($\delta_{\mathcal{T}} = 0$) by MOEA/D.

The proposed method mitigates the problem of duplicate Pareto objectives generated by MOEA/D through adaptive weight adjustment. Figure 6 and Table 3 demonstrate that the solutions generated by the proposed method possess the most uniform distribution on the Pareto front. Notably, the HV-based method in RE37 tends to produce solutions on the boundary of the Pareto front, which may not be desirable compared to the proposed method in certain applications.

③ Despite the general belief that hypervolume-based methods can generate diverse solutions in multi-objective optimization (MOO) (Auger et al., 2012; Guerreiro et al., 2020), our findings (see Table 5 in Appendix B.4 and Figure 9 in Appendix B.3) reveal that hypervolume indicators can be very similar, while the distribution of solutions can vary significantly. This highlights the need to utilize a novel indicator, as proposed, for measuring and optimizing the Pareto objectives for MOO. Furthermore, the proposed method is 4.5x faster than SMS-EMOA on RE37 (Table 2), since the proposed method only estimate and optimize the uniformity of a solution set only with minimal iterations. In most cases, the proposed method is optimized by MOEA/D under fixed preferences obtained from the neural PFL model.

## 6   CONCLUSIONS

This paper addresses a long-standing open problem in multiobjective evolutionary algorithms (MOEAs): the generation of a finite set of diverse/uniform Pareto objectives. It is the first paper to rigorously analyze the distribution of Pareto objectives, which sheds light on the understanding of solution generation in MOEAs. Building upon these analytical findings, the paper introduces a novel algorithm that achieves a uniform Pareto objective set through adaptive weight adjustment. In future research, we focus our attention on the acceleration of the optimization process and the application of the algorithm to large-scale MOO problems.

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

# A  METHODOLOGY

## A.1  DEFINITIONS AND MOO THEORY

As a preliminary, we introduce several topological concepts to describe basic geometric properties of a set $\mathcal{Y}$, from different perspectives. In Definition 4, we extend the concept of dominance to $\delta$-dominance. In Definitions 6 and 7, we introduce the $\delta$-packing and $\delta$-covering numbers of a set $\mathcal{Y}$, which are the maximal number of $\delta/2$-balls and minimal number of $\delta$-balls one needs, to pack in and cover $\mathcal{Y}$, respectively. They measure the metric entropies of $\mathcal{Y}$; See e.g., Vershynin (2018).

Definition 8 defines the diameter of $\mathcal{Y}$, which measures the supremum of the distances between pairs of points in $\mathcal{Y}$. Definition 9 gives a partition of $\mathcal{Y}$, given some point set $\mathcal{Y}_N$ in $\mathcal{Y}$. Briefly, any point in $\mathcal{Y}$ will be partitioned to the Voronoi cell that has the minimal distance to it. Intuitively, if points in $\mathcal{Y}_N$ are approximately evenly distributed, then they will induce Voronoi cells that have similar volumes to each other.

Finally, Hausdorff measure in Definition 10 introduces an adaptive and accurate way to measure the volumes of different sets in a multidimensional Euclidean space. For example, a curve has a trivial zero Borel measure in $\mathbb{R}^2$, yet the 1-dimensional Hausdorff measure is non-trivial and could measure its length. Likewise, for a spherical shell in $\mathbb{R}^3$ which has zero Borel measure, the corresponding 2-dimensional Hausdorff measures its surface area.

**Definition 4** ($\delta$-domination). *$\delta$-domination is an approximate Pareto solution defined with an error tolerance $\delta$. Specifically, a solution $x^{(a)}$ $\delta$-dominates (Zuluaga et al., 2016) $x^{(b)}$, if $\left( f(x^{(a)}) - \delta \right)$ dominates $f(x^{(b)})$.*

The definition of an aggregation function is adopted and modified from (Miettinen, 1999, Chapter 2.6), where it was originally referred to as the value function.

**Definition 5** (Aggregation function). *An aggregation function $g(\cdot) : \mathbb{R}^m \mapsto \mathbb{R}$. $g(\cdot)$ is a decreasing function, i.e., $g(x) < g(x')$, if $x_i < x_i'$, $\forall i \in [m]$.*

We have the following Lemmas (adopted and modified from (Miettinen, 1999, Thereom 2.6.2)) for an aggregation function,

**Lemma 2.** *Let $y^*$ be one of the optimal solution of $g(\cdot, \lambda)$, then $y^*$ is (weakly) Pareto optimal.*

**Lemma 3.** *Let $y^*$ be the only optimal solution of $g(\cdot, \lambda)$, then $y^*$ is Pareto optimal.*

**Definition 6** ($\delta$-packing, $\delta$-packing number). *A $\delta$-packing of a set $\mathcal{Y}$ is a collection of element $\{y^1, \ldots, y^n\}$ such that, $\rho(y^j, y^k) > \delta$ for all $j \neq k$. The $\delta$-packing number $N_{pack}(\delta, \mathcal{Y})$ is the largest cardinality among all $\delta$-packing. Here $\delta$ is called the packing distance.*

**Definition 7** ($\delta$-covering, $\delta$-covering number). *A $\delta$-covering of a set $\mathcal{Y}$ with respect to a metric $\rho$ is a set $\{\theta_1, \ldots, \theta_N\} \subset \mathcal{Y}$ such that for each $\theta \in \mathcal{Y}$, there exists some $i \in \{1, \ldots, N\}$ such that $\rho(\theta, \theta_i) \leq \delta$. The $\delta$-covering number $N_{cover}(\delta, \mathcal{Y})$ is the minimal cardinality among all $\delta$-coverings.*

**Definition 8** (Set diameter). *The diameter of any subset $\mathcal{Y} \subseteq \mathbb{R}^m$ is defined by*

$$\mathrm{diam}(\mathcal{Y}) = \max\{\rho(y_1, y_2) \mid y_1, y_2 \in \mathcal{Y}\}. \tag{10}$$

**Definition 9** (Voronoi cells (Okabe et al., 2009)). *The Voronoi cells $\{\mathcal{B}_1, \ldots, \mathcal{B}_N\}$ of a finite set $\mathcal{Y}_N \subseteq \mathcal{Y}$, where $\mathcal{Y}_N = \{y^{(1)}, \ldots, y^{(N)}\}$, is defined by,*

$$\mathcal{B}_i = \left\{ y \mid \min_{y \in \mathcal{Y}} \rho(y, y^{(i)}) \right\}, \quad \forall i \in [N]. \tag{11}$$

**Definition 10** (Hausdorff measure). *Consider the metric space $(\mathbb{R}^m, \rho)$ where $\rho(\cdot, \star)$ is the Euclidean distance. The $d$-dimensional Hausdorff measure of any Borel set $\mathcal{Y} \subseteq \mathbb{R}^m$ is*

$$\mathcal{H}_d(\mathcal{Y}) = \lim_{\delta \to 0} \inf_{\substack{\mathcal{Y} \subseteq \cup_{i=1}^{\infty} \mathcal{U}_i \\ \mathrm{diam}(\mathcal{U}_i) < \delta}} \left[ \sum_{i=1}^{\infty} \{\mathrm{diam}(\mathcal{U}_i)\}^d \right], \quad d \in (0, m).$$

We use Hausdorff measure to further define the Hausdorff dimension of a set. We first introduce the following Theorem 3 (Folland, 1999), which guarantees that, for any $\mathcal{Y}$, there exists at most one $d^\dagger \in \mathbb{R}$ which makes the $d^\dagger$-dimensional Hausdorff measure of $\mathcal{Y}$ both non-zero and finite. We call this $d^\dagger$ the Hausdorff dimension of $\mathcal{Y}$.

**Theorem 3.** *For any Borel set $\mathcal{Y} \subseteq \mathbb{R}^m$, suppose that for some $d^\dagger$, $0 < \mathcal{H}_{d^\dagger}(\mathcal{Y}) < \infty$. Then we have $\mathcal{H}_{d'}(\mathcal{Y}) = 0$ for any $d' < d^\dagger$ and $\mathcal{H}_{d'}(\mathcal{Y}) = \infty$ for any $d' > d^\dagger$.*

**Definition 11** (Hausdorff dimension). *We say the Hausdorff dimension of $\mathcal{Y}$ is $d^\dagger$ if and only if, $0 < \mathcal{H}_{d^\dagger}(\mathcal{Y}) < \infty$.*

## A.2  NOTATION TABLE

For clarity, we present a comprehensive list of symbols and notation in Table 4.

Table 4: The notation table.

| Notation | Meaning | Dimension |
|---|---|---|
| $N$ | Number of solutions. | - |
| $n$ | Dimension of a solution. | - |
| $m$ | Number of objectives. | - |
| $x$ | An MOO solution. | $n$ |
| $\mathcal{X}$ | The decision space. | $\mathbb{R}^n$ |
| $y, f(x)$ | The objective vector of $x$. | $m$ |
| $\mathcal{Y}$ | The objective space. | $\mathbb{R}^m$ |
| $\mathcal{Y}_N$ | A set of objectives, $\mathcal{Y}_N = \{y^{(1)}, \ldots, y^{(N)}\}$. | $m$ |
| $\mathcal{T}$ | The Pareto front. | $\mathbb{R}^m$ |
| $\delta_{\mathcal{T}}$ | The minimal separation distance of a set belong to $\mathcal{T}$ (Eq. 5). | |
| $\delta_v$ | The maximal diameter of all Voronoi cells. (Eq. 18). | |
| $\lambda$ | The preference vector. | $m$ |
| $\theta(\lambda), \theta$ | The angular parameterization of vector $\lambda$ (Defined in Section A.3). | $m-1$ |
| $g^{\mathrm{alg}}(\cdot\|\lambda)$ | The multiobjective aggregation function. | $\mathbb{R}^m \mapsto \mathbb{R}$ |
| $h(\cdot)$ | The function maps a preference to a Pareto solution. $h(\cdot) : \Omega \mapsto \mathbb{R}^m$. | |
| $h_\phi(\cdot)$ | The PFL model. $h_\phi(\cdot) : \left[0, \frac{\pi}{2}\right]^{m-1} \mapsto \mathbb{R}^m$ | |
| $\mathcal{H}_d(\cdot)$ | Hausdorff dimension function . | |

## A.3  THE RELATION BETWEEN PREFERENCE ANGLE $\theta(\lambda)$ AND PREFERENCE $\lambda$

Given $\theta(\lambda) \in [0, \frac{\pi}{2}]^{m-1}$ as a parameter representation of $\lambda \in \Delta_{m-1}$, the preference angle and the corresponding preference vector can be converted using the following equations:

$$
\begin{cases}
\lambda_1 = \sqrt{\cos(\theta_1)} \\
\lambda_2 = \sqrt{\sin(\theta_1)\cos(\theta_2)} \\
\quad\vdots \\
\lambda_m = \sqrt{\sin(\theta_1)\sin(\theta_2)\ldots\sin(\theta_{m-1})}.
\end{cases}
\tag{12}
$$

For a given preference vector $\lambda$, computing the preference angle can be achieved by solving Equations (12).

## A.4  ALGORITHM

The total algorithms run as Algorithm 1 and 2. The proposed algorithm mainly adopt the MOEA/D framework. For simplicity, we use the Simulated Binary Crossover (SBX) (Deb et al., 1995) and poly-nominal mutation mutation operators (Deb & Deb, 2014). It is possible to use more advanced MOEA/D framework, e.g., MOEA/D with differential evolution (Tan et al., 2012), which is left as future works.

As summarized by Algorithm 1, the proposed approach dynamically adjusts the preference angles on the estimated Pareto front learned by the current objectives. The updated preferences are then set as the new preferences for the MOEA/D algorithm.

---

**Algorithm 1** Training of PFL and preference adjustment.

---

**Input:** Preference angle set $\Theta_N$ and objective set $\mathcal{Y}_N$ from MOEA/D.

# Training the PFL model.

**for** *i=1:$N_{pfl}$* **do**

$\quad | \quad \phi \leftarrow \phi - \tilde{\eta}\nabla_\phi l_{\text{pfl}}$.

**end**

# Solving the maximal separation problem at the estimated Pareto front (Problem (6)) by gradient ascent.

**for** *i=1:$N_{opt}$* **do**

$\quad | \quad \Theta_N \leftarrow \text{Proj}(\Theta_N + \eta\nabla_{\Theta_N}\hat{\delta}_{\hat{\mathcal{T}}})$.

**end**

**Output:** The updated preference angles $\{\theta^{(1)}, \ldots, \theta^{(N)}\}$.

---

---

**Algorithm 2** MOEA/D with uniform adaptive preference adjustment (MOEA/D-UAWA).

---

**Input: Initial $N$ preference $\Lambda_N$ : $\Lambda_N = \left\{\lambda^{(1)}, \ldots, \lambda^{(N)}\right\}$ by (Das & Dennis, 1998), the initial solution set $\mathcal{X}_N$ : $\mathcal{X}_N = \left\{x^{(1)}, \ldots, x^{(N)}\right\}$, the MOO objective function $f(\cdot)$.**

**for** *k=1:K* **do**

$\quad$ **for** *i=1:$N_{inner}$* **do**

$\quad\quad$ # Step 1. Evolutionary algorithm.

$\quad\quad$ **for** *j=1:N* **do**

$\quad\quad\quad$ # Generate a crossover solution from neighbourhoods of $x^{(j)}$ using SBX operator.

$\quad\quad\quad$ $x^{(j)} \leftarrow \text{SBX}(x^{(j_1)}, x^{(j_2)})$, where $x^{(j_1)}, x^{(j_2)}$ are selected randomly from the neighborhood set of $x^{(j)}$.

$\quad\quad\quad$ # Mutation the solution by the polynomial mutation operator.

$\quad\quad\quad$ $x^{(j)} \leftarrow \text{Mutate}(x^{(j)})$.

$\quad\quad$ **end**

$\quad\quad$ # Update the solution each sub-problems by elites.

$\quad\quad$ **for** *j=1:N* **do**

$\quad\quad\quad$ $x^{(j)} \leftarrow \arg\min_{i\in B(j)\cup\{x^{(j)}\}} g^{\text{mtche}}(f(x^i), \lambda^{(j)})$. $B(j)$ is the neighborhood index set (Zhang & Li, 2007) of solution $x^{(j)}$.

$\quad\quad$ **end**

$\quad$ **end**

$\quad$ # Step 2. Preference adjustment.

$\quad$ Calculate the preference angle set $\Theta_N$ from preferences by Equation (12).

$\quad$ $\mathcal{Y}_N = f \circ \mathcal{X}_N$.

$\quad$ $\Theta_N \leftarrow \text{Algorithm1}(\Theta_N, \mathcal{Y}_N)$.

$\quad$ Update the preference vector $\lambda^{(1)}, \ldots, \lambda^{(N)}$ by Equation (12).

**end**

---

We briefly analyze the running complexity of the proposed method. The main complexity is inherited from MOEA/D (Zhang & Li, 2007). The addition of the PFL model training and the uniformity optimization introduces two additional parts.

Training the PFL model is a standard supervised learning problem, hence the complexity is proportional to the number of objectives $m$ and sample numbers $N$. The overall complexity is $\mathcal{O}(mN \cdot N_{\text{pfl}})$.

The uniformity optimization involves calculating the lower (or upper) triangular matrix of an adjacency matrix, which has a complexity of $\mathcal{O}(m \cdot \frac{N(N-1)}{2}) = \mathcal{O}(mN^2)$. Therefore, the total complexity of the optimization process is $\mathcal{O}(mN_{\text{opt}} \cdot \frac{mN(N-1)}{2}) = \mathcal{O}(mN^2 \cdot N_{\text{opt}})$.

Practically, training time of the PFL and preference adjustment is less than `1s`, which is neglectable compared with MOEAs. The calculation of the adjacency matrix and the MOEA/D algorithm can be executed in parallel, which can further improve the efficiency of the overall running time.

## B   EXPERIMENTS DETAILS

### B.1   COMPARISON METHODS

We give a detailed elaboration of the comparison methods used in the experiments as follows. The code for the proposed method will be made publicly available upon acceptance.

1. The vanilla **MOEA/D** method (Zhang & Li, 2007) employs diverse distributed preference vectors to explore a diverse Pareto solution set. However, the uniformity observed in the preference space may not lead to uniformity in the objective space, resulting in a coarse level of solution diversity sought by MOEA/D.

2. The non-dominated sorting genetic algorithm ( **NSGA2** ) (Deb et al., 2002a), which applies the concepts of non-dominated sorting and crowding distance to select the most elite solutions that cover the diverse space of the Pareto front. To maintain solution diversity in the population, the selection process uses tournament selection and crowding distance calculation. The code directly follows the pymoo library.

3. The MOEA/D with adaptive weight adjustment ( **MOEA/D-AWA** ) (Qi et al., 2014). MOEA/D-AWA is an improvement over the vanilla MOEA/D, which aims to improve the replaces the most crowded solution with the most sparse solution. A detailed comparison of MOEA/D-AWA and the proposed method can be found in Appendix B.5.

   Since the source code for their implementation was not publicly available, we implemented it by ourselves. In the original implementation of MOEA/D-AWA, they maintain an external population (EP). However, as the modern MOEA frameworks (e.g., pymoo, platemo) are no longer dependent on EP, we employ a neural network surrogate model to predict the most sparse solution (Qi et al., 2014).

4. The Pareto adaptive weight (**PaLam**) method (Siwei et al., 2011) approximates the Pareto front using a simple math mode $y_1^p + y_2^p = 1$ and generates uniform Pareto objectives by utilizing the hypervolume indicator (Guerreiro et al., 2020).

   Since real-world problems often exhibit complex Pareto fronts, to ensure fairness, we employ a neural model for training to predict the true Pareto front instead of relying on a simple mathematical model. We use the code in https://github.com/timodeist/multi_objective_learning to develop a new gradient-based algorithm for pa$\lambda$ to achieve fast optimization for the `HV` indicator predicted by neural networks. Our improved version of the vanilla pa$\lambda$ significantly outperforms its original implementation.

5. The **SMS-EMOA** (Beume et al., 2007) method, which uses the hypervolume indicator as the guidance for the multiobjective evolutionary optimizations. The code for SMS-EMOA directly follows the pymoo library.

### B.2   METRICS

In order to measure the performance of our proposed method, we employ the following metrics to measure uniformity and the solution quality. The up-arrow ($\uparrow$)/down-arrow($\downarrow$) signifies that a higher/lower value of this indicator is preferable.

1. The `hypervolume` ($\uparrow$) (`HV`) indicator (Guerreiro et al., 2020), which serves as a measure of the convergence (the distance to the Pareto front) and only a *coarse* measure of the sparsity/uniformity level of a solution set.

2. The `sparsity` ($\downarrow$) indicator (Xu et al., 2020), which measures the sum of distance of a set of solutions in the non-dominated sorting order (Deb et al., 2002a).

3. The `spacing` ($\downarrow$) indicator (Schott, 1995), which measures the uniformity of a set of solutions. It is quantified as the standard deviation of the distance set $\{d_1, \ldots, d_N\}$.

$$\texttt{spacing}(\mathcal{Y}_N) = \text{std}(d_1, \ldots, d_N), \tag{13}$$

where $d_i = \min_{j \in [m], j \neq i} \rho(y^{(i)}, y^{(j)})$, serving as the minimal distance from solution $y^i$ to the rest of objectives.

4. The $\delta_\mathcal{T}$ (↑) and $\tilde{\delta}_\mathcal{T}$ (↑) indicators represent the (soft) minimal distances among different solutions within a solution set.

$$\tilde{\delta}_\mathcal{T} = -\frac{1}{K} \log \sum_{i \neq j} \exp\left(-K \cdot \rho(y^{(i)}, y^{(j)})\right). \tag{14}$$

A large $\delta_\mathcal{T}/\tilde{\delta}_\mathcal{T}$ indicator means that, any different solution pairs are far away from each other. Noted that, it is possible that $\tilde{\delta}_\mathcal{T} < 0$. $K$ is a positive constant.

### B.3 LEARNING PROCESS OF THE PROPOSED METHOD

In the subsection, we show the process of the proposed method find the uniform Pareto objectives. We take two problems as an example, the two-objective ZDT1 problem and the three-objective DTLZ2.

**(ZDT1 problem.)** We first investigate the optimization process of $\delta_\mathcal{T}$ (MMD) on a simple ZDT1 problem. Before the first preference adjustment, the MMD indicator is around -0.15, and after the first round of adjustment, this indicator is optimized to -0.21 in the estimated Pareto front. In the third adjustment, the MMD indicator is always around -0.2, which indicates that the solutions are distributed uniformally.

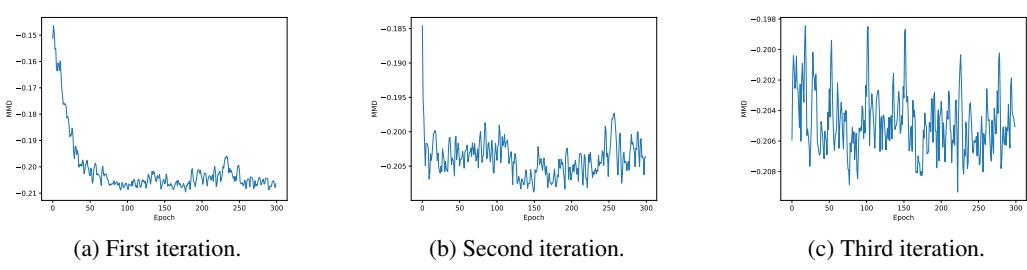

(a) First iteration.    (b) Second iteration.    (c) Third iteration.

Figure 7: The $\delta_\mathcal{T}$ (MMD) optimization curve in ZDT1.

Figure 8 shows the MSE loss of the PFL model during optimization. It is evident that the training loss is optimized to zero after only a few epochs. Additionally, we observe that the model adapts well to new solutions. When the solutions are updated to a new position, the initial loss remains very low.

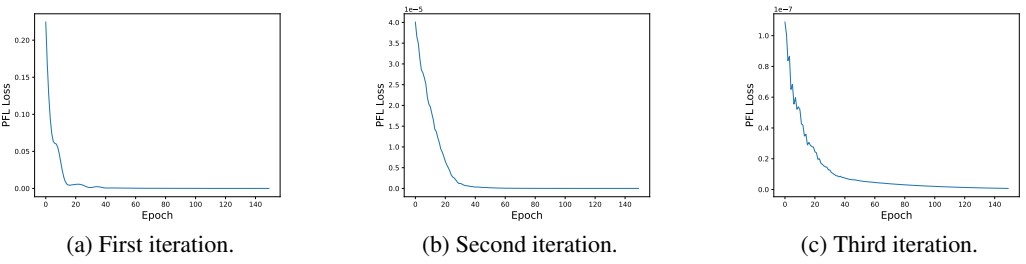

(a) First iteration.    (b) Second iteration.    (c) Third iteration.

Figure 8: The PFL loss curve in ZDT1.

**(The DTLZ2 problem.)** We plot the MMD and hypervolume on the DTLZ2 problem in Figure 9. Figure 9 clearly demonstrates that the MMD indicator steadily increases with each iteration, indicating a progressive improvement in the uniformity of the solutions. Notably, our findings challenge the conventional belief that the hypervolume indicator measures the uniformity of a Pareto set. We discovered that the hypervolume indicators can be very similar while the distribution of Pareto

objectives are very different. This suggests that the hypervolume indicator alone is an imprecise measure of solution uniformity.

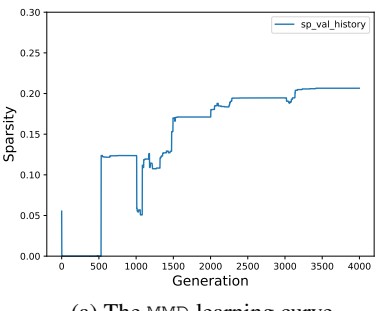

(a) The `MMD` learning curve

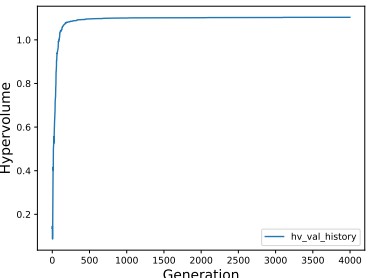

(b) The `hypervolume` learning curve.

Figure 9: Learning curves on the DTLZ2 problem.

### B.4 RESULTS ON ALL PROBLEMS

In this subsection, we visualize the results on all problems through Figure 10 to 17. Those figures depict the outcomes of MOEA/D, SMS-EMOA, and MOEA/D-AWA. To conserve space, the results on NSGA2 and PaLam are excluded since NSGA2 is less relevant compared to our method, and PaLam has been significantly modified, yielding similar results to our proposed method with only a different uniformity indicator. The numerical results are shown in Table 5.

Regarding two-objective problems, depicted in Figures 10-14. We summarize the key findings from the experiments in the rest of this subsection.

**(Neighbour distances are equal in two-objective problems).** For the standard ZDT1 and ZDT2 problems, which have a Pareto front ranging from zero to one, the Pareto objectives optimized by MOEA/D are not uniformly distributed. However, our method ensures that the distance between adjacent solutions is equal, indicating a more uniform distribution. For MOEA/D, when considering a convex-shaped Pareto front like ZDT1, solutions tend to be denser in the middle. Conversely, for a concave-shaped Pareto front like ZDT2, solutions are denser towards the margins. However, the proposed method's Pareto objective distribution remains unaffected by the shape of the true Pareto front.

**(The proposed method is robust to objectives with different scales).** When function ranges differ in scale (see Figure 13-14), the uniformity of pure MOEA/D worsens. In this scenario, achieving preference uniformity does not guarantee objective uniformity. Solutions become even sparser in the upper-left region of the objective space. In contrast, the proposed method consistently generates uniform Pareto objectives. Hypervolume-based methods remain unaffected when function objectives have different scales. However, objectives produced by hypervolume-based methods are not strictly uniform, and hypervolume-based methods are typically slower.

**(HV is not a accurate uniformity indicator).** The hypervolume indicator only provides an approximate measure of solution diversity. Table 5 illustrates that similar hypervolume indicators can correspond to significantly different solution distributions. In the case of hypervolume-based methods (PaLam or SMS-EMOA), the largest hypervolume indicator does not necessarily lead to the most uniform objectives for most problems.

**(Preference uniformity induces solution uniformity in DTLZ1).** As mentioned in Remark 1, the Pareto front of DTLZ1 can be represented as an affine transformation of the 2-D simplex $\Delta_2$ ($h(x) = \frac{1}{2}x$). In this paper, this is the only scenario where uniform preferences result in uniformly distributed Pareto objectives. Hence, MOEA/D with uniform preferences performs exceptionally well on the DTLZ1 task. The maximal manifold separation indicator, $\delta_{\mathcal{T}}$, outperforms all other

methods, and the proposed method achieves a value of $\delta_{\mathcal{T}} = 0.099$, which is very close to that of MOEA/D and significantly outperforms other methods.

**(Results on the difficult RE37).** Finally, we present the results for the challenging three-objective real-world problem RE37. One of the difficulties of this problem is the presence of many preferences within the preference simplex $\Delta_2$ that do not intersect with the Pareto front (PF). The pure MOEA/D algorithm produces numerous duplicate solutions when preferences do not intersect with the Pareto front (PF), resulting in wasted resources and poor solution diversity. The hypervolume-based method SMS-EMOA exhibits an interesting phenomenon: it mainly focuses on the marginal region of the PF, which may not always meet user demands. In contrast, the proposed method is the only method that generates uniform objectives that cover the entire PF.

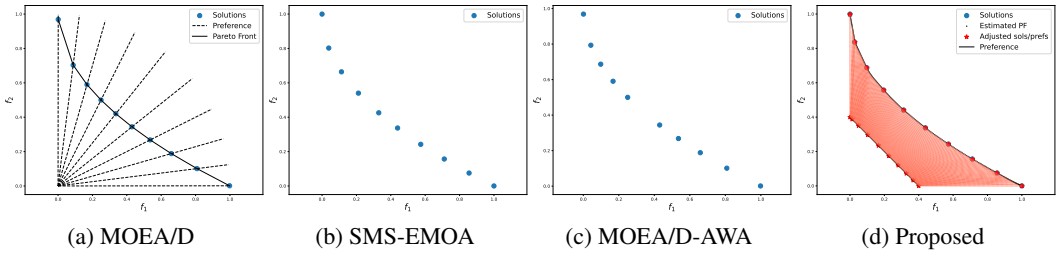

Figure 10: Results on ZDT1.

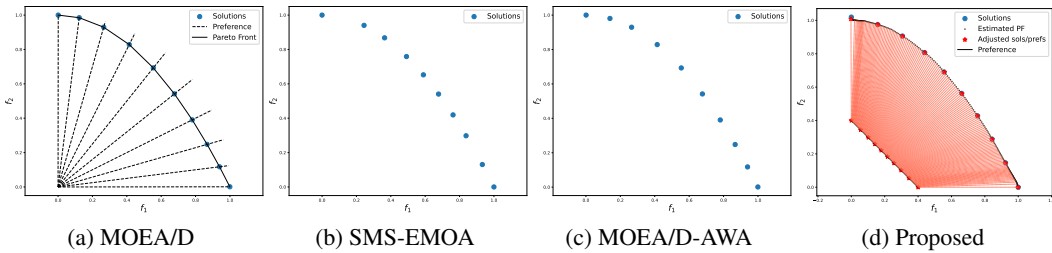

Figure 11: Results on ZDT2.

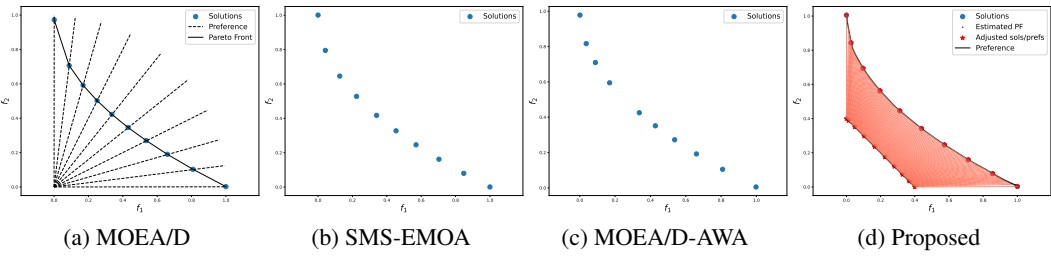

Figure 12: Results on ZDT4.

## B.5 DIFFERENCE OF THE PROPOSED MOEA/D-UAWA AND MOEA/D-AWA

The preference adjustment process of MOEA/D-AWA (Qi et al., 2014) is demonstrated in Figure 18. During each update, the algorithm eliminates the most crowded objective (depicted as a red dot in Figure 7) and adds the most sparse objective (represented by a green dot). The resulting preference is indicated by a green star. The sparsity measure is determined using Equation (4) as outlined in the original publication (Qi et al., 2014). From Figure 18, it is evident that the initial objectives obtained

Table 5: Results on all problems averaged on five random seeds, with the optimal indicators among all problems highlighted in bold.

| Problem | Method | HV ($\uparrow$) | Spacing ($\downarrow$) | Sparsity ($\downarrow$) | $\delta_{\mathcal{T}}$ ($\uparrow$) | $\tilde{\delta}_{\mathcal{T}}$ ($\uparrow$) |
|---------|--------|-----|---------|----------|-----------------|-----------------|
| ZDT1 | NSGA2 | 1.036 | 0.072 | 0.033 | 0.032 | -0.039 |
| | SMS-EMOA | 1.053 | 0.021 | 0.027 | 0.141 | 0.013 |
| | MOEA/D | 1.050 | 0.050 | 0.028 | 0.117 | -0.005 |
| | MOEA/D-AWA | 1.051 | 0.031 | 0.027 | 0.120 | 0 |
| | PaLam | **1.054** | 0.018 | 0.027 | 0.148 | 0.015 |
| | Proposed | 1.053 | **0.002** | **0.027** | **0.161** | **0.018** |
| ZDT2 | NSGA2 | 0.708 | 0.066 | 0.035 | 0.018 | -0.048 |
| | SMS-EMOA | **0.726** | 0.041 | 0.029 | 0.130 | 0.006 |
| | MOEA/D | 0.722 | 0.025 | 0.028 | 0.123 | 0.011 |
| | MOEA/D-AWA | 0.722 | 0.023 | 0.027 | 0.131 | 0.012 |
| | PaLam | 0.722 | 0.025 | 0.028 | 0.146 | 0.014 |
| | Proposed | 0.724 | **0.001** | **0.027** | **0.161** | **0.018** |
| ZDT4 | NSGA2 | 1.041 | 0.053 | 0.032 | 0.055 | -0.023 |
| | SMS-EMOA | **1.052** | 0.024 | 0.028 | 0.133 | 0.012 |
| | MOEA/D | 1.046 | 0.051 | 0.028 | 0.118 | -0.004 |
| | MOEA/D-AWA | 1.040 | 0.030 | 0.028 | 0.116 | 0.001 |
| | PaLam | 1.049 | 0.015 | 0.027 | **0.146** | **0.016** |
| | Proposed | 1.050 | **0.012** | **0.027** | 0.139 | 0.015 |
| RE21 | NSGA2 | 1.226 | 0.066 | 0.022 | 0.024 | -0.063 |
| | SMS-EMOA | 1.252 | 0.028 | 0.017 | 0.095 | -0.028 |
| | MOEA/D | 1.246 | 0.086 | 0.024 | 0.072 | -0.058 |
| | MOEA/D-AWA | 1.250 | 0.028 | 0.017 | 0.088 | -0.031 |
| | PaLam | **1.252** | 0.025 | 0.017 | 0.101 | -0.027 |
| | Proposed | 1.252 | **0.002** | **0.016** | **0.123** | **-0.020** |
| RE22 | NSGA2 | 1.155 | 0.04 | 0.022 | 0.033 | -0.055 |
| | SMS-EMOA | **1.191** | 0.021 | 0.018 | 0.092 | -0.025 |
| | MOEA/D | 1.186 | 0.046 | 0.02 | 0.06 | -0.048 |
| | MOEA/D-AWA | 1.189 | 0.025 | 0.017 | 0.092 | -0.028 |
| | PaLam | 1.188 | 0.014 | 0.018 | 0.106 | **-0.018** |
| | Proposed | 1.19 | **0.002** | **0.017** | **0.125** | -0.019 |
| DTLZ1 | NSGA2 | 1.692 | 0.022 | **0.001** | 0.008 | -0.194 |
| | SMS-EMOA | **1.697** | 0.011 | 0.001 | 0.058 | -0.172 |
| | MOEA/D | 1.696 | **0.000** | 0.003 | **0.100** | -0.107 |
| | MOEA/D-AWA | 1.696 | 0.010 | 0.003 | 0.058 | **-0.169** |
| | PaLam | 1.697 | 0.019 | 0.001 | 0.030 | -0.172 |
| | Proposed | 1.697 | 0.001 | 0.003 | 0.099 | -0.169 |
| DTLZ2 | NSGA2 | 1.033 | 0.070 | **0.004** | 0.010 | -0.126 |
| | SMS-EMOA | **1.117** | 0.053 | 0.010 | 0.087 | -0.073 |
| | MOEA/D | 1.100 | 0.045 | 0.008 | 0.164 | -0.019 |
| | MOEA/D-AWA | 1.101 | 0.051 | 0.008 | 0.080 | -0.035 |
| | PaLam | 1.106 | 0.059 | 0.006 | 0.013 | -0.081 |
| | Proposed | 1.104 | **0.023** | 0.011 | **0.205** | **-0.007** |
| RE37 | NSGA2 | 1.051 | 0.069 | 0.005 | 0.013 | -0.140 |
| | SMS-EMOA | **1.114** | **0.041** | 0.005 | 0.029 | -0.128 |
| | MOEA/D | 1.052 | 0.072 | 0.013 | 0 | -0.204 |
| | MOEA/D-AWA | 1.091 | 0.078 | 0.009 | 0.001 | -0.177 |
| | PaLam | 1.112 | 0.076 | 0.006 | 0 | -0.174 |
| | Proposed | 1.110 | 0.045 | **0.005** | **0.040** | **-0.086** |

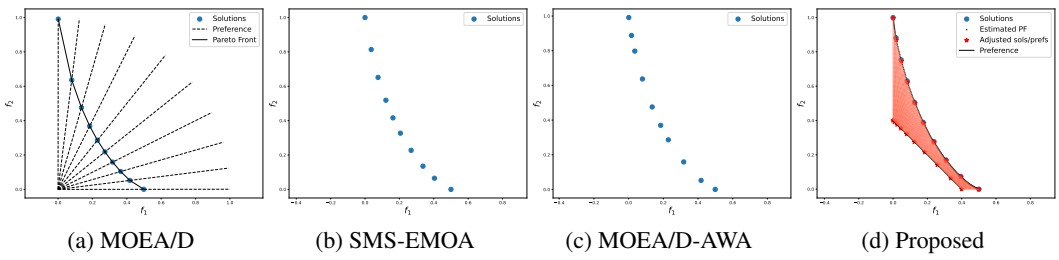

(a) MOEA/D      (b) SMS-EMOA      (c) MOEA/D-AWA      (d) Proposed

Figure 13: Results on RE21.

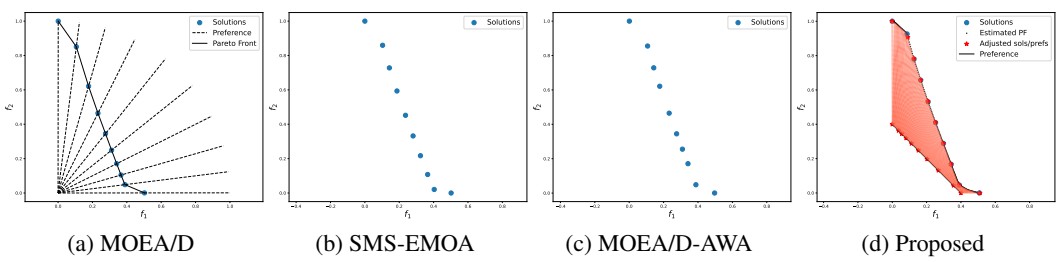

(a) MOEA/D      (b) SMS-EMOA      (c) MOEA/D-AWA      (d) Proposed

Figure 14: Results on RE22.

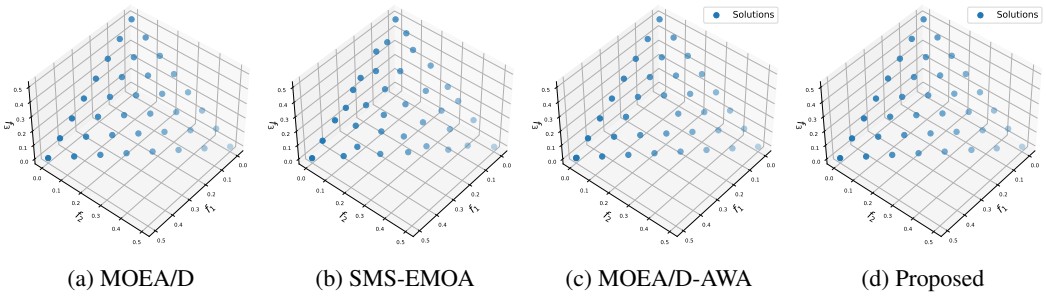

(a) MOEA/D      (b) SMS-EMOA      (c) MOEA/D-AWA      (d) Proposed

Figure 15: Results on DTLZ1.

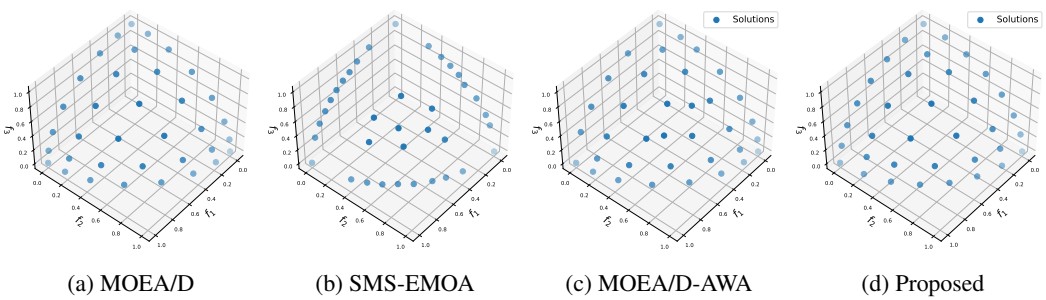

(a) MOEA/D      (b) SMS-EMOA      (c) MOEA/D-AWA      (d) Proposed

Figure 16: Results on DTLZ2.

by MOEA/D-mtche (MOEA/D with modified Tchebycheff aggregation function) are unevenly distributed on the (surrogate) Pareto front. By contrast, MOEA/D-AWA successfully eliminates the most crowded objective in the upper-right region of the DTLZ2 problem, adding a new objective at the center of the Pareto front.

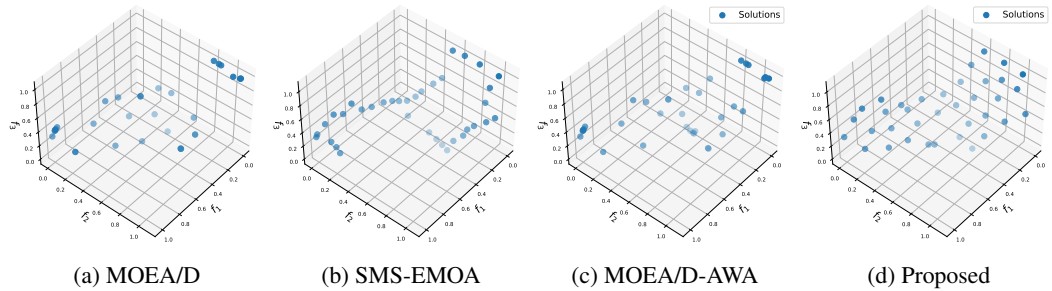



(a) MOEA/D       (b) SMS-EMOA      (c) MOEA/D-AWA      (d) Proposed

Figure 17: Results on RE37.



However, the aforementioned strategy is heuristic in nature, lacking a guarantee of achieving optimal solutions during the final adjustment phase. Another difference is MOEA/D-AWA compared with the proposed method is that it only remove and add one solution for each preference adjustment, which make it less efficient compared with the proposed method.

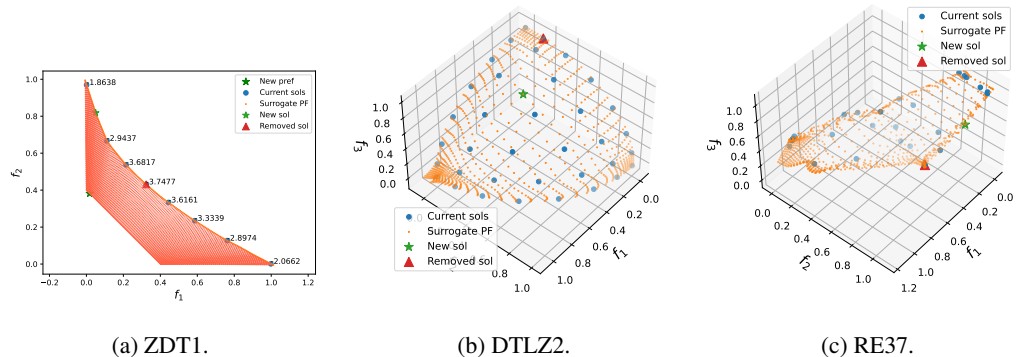



(a) ZDT1.       (b) DTLZ2.       (c) RE37.

Figure 18: Preference adjustment in MOEA/D-AWA.



### B.6 COMPARISON WITH GRADIENT-BASED MOO

We take the ZDT4 problem as an example, which has the form of,

$$
\begin{cases}
f_1(x) = x \\
g(x) = 1 + 10(n-1) + \sum_{i=2}^{n}(x_i^2 - 10\cos(4\pi x_i)) \\
h(f_1, g) = 1 - \sqrt{f_1/g} \\
f_2(x) = g(x)h(f_1(x), g(x)) \qquad\qquad 0 \le x_1 \le 1, \\
-10 \le x_i \le 10, \quad i = 2, \dots, n.
\end{cases}
\tag{15}
$$

It is clear that due to the term $\sum_{i=2}^{n}(x_i^2 - 10\cos(4\pi x_i))$, the objective function has plenty of locally optimas. Simply using gradient methods fails on this problem as shown in Figure 19.

## C THEORETICAL RESULTS

In this section, we provide some theoretical results of the benefits of the uniform Pareto objectives. Section C.1 provides the preliminary tools of risk analysis which is served for Section C.2. Section C.2 proves the main Theorem 2 in the main paper. The other sections provide the missing proofs in Section 4.1 and 4.2.

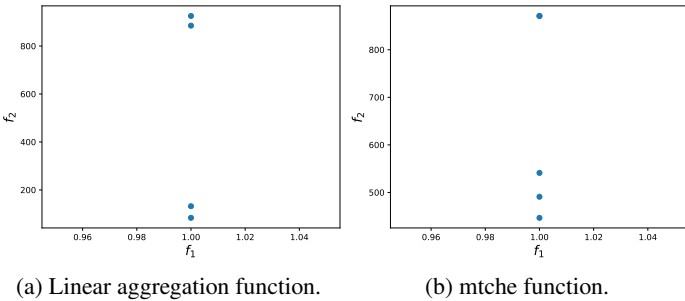

(a) Linear aggregation function.      (b) mtche function.

Figure 19: Gradient-based MOO on ZDT4.

## C.1 RISK DECOMPOSITION

The excess risk of the prediction model, i.e., $R(\hat{h})$, can be decomposed in four parts according to Telgarsky (2021),

$$R(\hat{h}) \leq \underbrace{\left| R(\hat{h}) - \hat{R}(\hat{h}) \right|}_{\epsilon_1} + \underbrace{\hat{R}(\hat{h}) - \hat{R}(\overline{h})}_{\epsilon_2} + \underbrace{\left| \hat{R}(\overline{h}) - R(\overline{h}) \right|}_{\epsilon_3} + \underbrace{R(\overline{h})}_{\epsilon_4}; \tag{16}$$

Here $R(\cdot)$ denotes the true risk and $\hat{R}(\cdot)$ denotes the empirical risk with $n$ samples, $\hat{h}$ denotes the function achieved by SGD and $\overline{h}$ is some low-complexity approximation of $h_*(\cdot)$ in a function class $\mathcal{F}$ of all possible neural-networks predictors.

The second term $\epsilon_2$ is the optimization loss, which can be optimized to global optimal in a time polynomially in the network depth and input size according to Allen-Zhu et al. (2019, Th. 1). The forth term $\epsilon_4$ is the approximation error, both of which can be small due to the universal approximation theorem with an appropriately specified $\mathcal{F}$; See e.g., Telgarsky (2021, Preface).

In the next section, we give the bound of the first and third term $\epsilon_1$ and $\epsilon_3$ through controlling the more general generalization error. In particular, let $\tilde{h} = \overline{h}$ or $\hat{h}$, then $\epsilon_1$ or $\epsilon_3$ can be expressed by the corresponding generalization error $\tilde{\epsilon} = |R(\tilde{h}) - \hat{R}(\hat{h})|$.

## C.2 PROOF OF THEOREM 2

*Proof.* We firstly decompose the error $\tilde{\epsilon}$ into two parts, $\varepsilon_1$ and $\varepsilon_2$, which can be formulated as,

$$\tilde{\epsilon} = \underbrace{\int_{\mathcal{T}} \left( \left\| (\tilde{h} - h_*) \circ h_*^{-1}(y) \right\| \right)^2 dy}_{R(\tilde{h})} - \underbrace{\frac{1}{N} \sum_{i=1}^{N} \left( \left\| (\tilde{h} - h_*) \circ h_*^{-1}(y_i) \right\| \right)^2}_{\hat{R}(\tilde{h})},$$

$$\leq \underbrace{\sum_{i=1}^{N} \left( \int_{\mathcal{B}_i} \left( \left( \left\| (\tilde{h} - h_*) \circ h_*^{-1}(y) \right\| \right)^2 - \left( \left\| (\tilde{h} - h_*) \circ h_*^{-1}(y_i) \right\| \right)^2 \right) dy \right)}_{\varepsilon_1} \tag{17}$$

$$+ \underbrace{\sum_{i=1}^{N} \left( \frac{\mathcal{H}_{m-1}(\mathcal{B}_i)}{\mathcal{H}_{m-1}(\mathcal{T})} - \frac{1}{N} \right) c_i}_{\varepsilon_2},$$

where $h_*(\cdot)$ is the true mapping function from a preference angle to Pareto solution, $h_*^{-1}(\cdot)$ is the inverse function of $h_*(\cdot)$ and $c_i = \left( \left\| (\tilde{h} - h_*) \circ h_*^{-1}(y_i) \right\| \right)^2 \leq A^2$. We use the notation $c$ to denote the maximal value of $c_i$, where $0 < c = \max_{i \in [N]}\{c_i\} \leq A^2$. In the second line in Equation (17), we use the notation $\mathcal{B}_i$ to denote the Voronoi cell of point $y_i$, where $\mathcal{B}_i = \{y \mid \min_{y \in \mathcal{T}} \rho(y, y_i)\}$.

The distance function $\rho(\cdot, \cdot)$ remain as the $l_2$ norm. We use $\delta_i$ to denote the diameter of set $\mathcal{B}_i$, where the formal definition of a diameter is provided in Equation (10). Similarly, the maximal diameter $\delta_v$ of all cells is defined as,

$$\delta_v = \max_{i \in [N]} \{\delta_i\}. \tag{18}$$

The proof primarily consists of two steps, where we separately bound $\varepsilon_1$ and $\varepsilon_2$.

**(The bound of $\varepsilon_1$)** We first show that $\varepsilon_1$ can be bounded by the maximal diameter $\delta$ up to a constant.

$$\sum_{i=1}^{N} \int_{\mathcal{B}_i} \left( \left\| (\tilde{h} - h_*) \circ h_*^{-1}(y) \right\| \right)^2 - \left( \left\| (\tilde{h} - h_*) \circ h_*^{-1}(y_i) \right\| \right)^2 dy$$

$$= \sum_{i=1}^{N} \int_{\mathcal{B}_i} \left( \left\| (\tilde{h} - h_*) \circ h_*^{-1}(y) \right\| - \left\| (\tilde{h} - h_*) \circ h_*^{-1}(y_i) \right\| \right) \left( \left\| (\tilde{h} - h_*) \circ h_*^{-1}(y) \right\| + \left\| (\tilde{h} - h_*) \circ h_*^{-1}(y_i) \right\| \right) dy$$

$$\leq \sum_{i=1}^{N} \int_{\mathcal{B}_i} \left( \left\| (\tilde{h} - h_*) \circ h_*^{-1}(y) - (\tilde{h} - h_*) \circ h_*^{-1}(y_i) \right\| \right) \left( \left\| (\tilde{h} - h_*) \circ h_*^{-1}(y) \right\| + \left\| (\tilde{h} - h_*) \circ h_*^{-1}(y_i) \right\| \right) dy$$

(By function smoothness and upper bound)

$$\leq \sum_{i=1}^{N} \int_{\mathcal{B}_i} 2AA'LL' \left\| y - y_i \right\| dy$$

$$\leq \sum_{i=1}^{N} 2\mathcal{H}_{m-1}(\mathcal{B}_i) AA'LL'\delta_v$$

$$\leq 2\mathcal{H}_{m-1}(\mathcal{T}) AA'LL'\delta_v. \tag{19}$$

**(The bound of $\varepsilon_2$)** To bound $\varepsilon_2$, we introduce several distribution functions as summarized below.

- $\mathcal{U} = \mathrm{Unif}(\mathcal{T})$ denotes the uniform distribution over the Pareto front $\mathcal{T}$.

- $\widetilde{\mathcal{Y}}_N$ represents the category distribution over the set $\{y^{(1)}, \ldots, y^{(N)}\}$, where each discrete point has a probability of $\frac{1}{N}$:

- $S_N$ is any distribution, that satifies the following properties. (1) $\int_{\mathcal{B}_i} p_{S_N}(y) dy = \frac{1}{N}$ (2) $|\partial p_{S_N}(y)/\partial y|$ is zero at boundary and is bounded at other place, and (3) almost surely, the pointwise density of $S_N$ is large or smaller than the corresponding pointwise density of $\mathcal{U}$, at each $\mathcal{B}_i$.

With the above distributions, we can bound $\varepsilon_2$ by the following derivation,

$$\sum_{i=1}^{N} \left( \frac{\mathcal{H}_{m-1}(\mathcal{B}_i)}{\mathcal{H}_{m-1}(\mathcal{T})} - \frac{1}{N} \right) c_i \leq \sum_{i=1}^{N} \left| \frac{\mathcal{H}_{m-1}(\mathcal{B}_i)}{\mathcal{H}_{m-1}(\mathcal{T})} - \frac{1}{N} \right| A^2$$

$$= CA^2 \mathrm{TV}(\mathcal{U}, S_N)$$

$$\leq CA^2 \sqrt{\mathcal{W}_1(\mathcal{U}, S_N)} \tag{20}$$

$$\leq CA^2 \sqrt{\mathcal{W}_1(\mathcal{U}, \widetilde{\mathcal{Y}}_N) + \mathcal{W}_1(\widetilde{\mathcal{Y}}_N, S_N)}$$

$$\leq CA^2 \sqrt{\mathcal{W}_1(\mathcal{U}, \widetilde{\mathcal{Y}}_N) + \delta_v}.$$

Here, $\mathcal{W}_1$ is the Wasserstein distance function. The second line is from the definition of the total variance (TV) distance. The third line is an adaptation of (Chae & Walker, 2020, Theorem 2.1) with

$\alpha = 1$ therein. As shown by (Chae & Walker, 2020, Theorem 2.1), $C > 0$ is determined by the Sobolev norms of $\mathcal{U}$ and $S_N$, which can be regarded as a universal constant, since $S_N$ has a smooth density function and $\mathcal{T}$ is compact. The quantity $\mathcal{W}_1(\widetilde{\mathcal{Y}}_N, S_N)$ can be bounded by the following expressions,

$$
\begin{aligned}
\mathcal{W}_1(\widetilde{\mathcal{Y}}_N, S_N) &= \inf_{\gamma \in \Gamma} \int_{\mathcal{T} \times \mathcal{T}} |y - y'| \, \gamma(y, y') dy dy' \\
&\leq \inf_{\gamma \in \Gamma} \sum_{i=1}^N \int_{\mathcal{T}} |y - y_i| \, \gamma(y, y_i) dy \\
&\leq \delta_v \inf_{\gamma \in \Gamma} \sum_{i=1}^N \underbrace{\int_{\mathcal{T}} \gamma(y, y_i) dy}_{=1/N} = \delta_v.
\end{aligned}
\tag{21}
$$

Here $\Gamma$ is the set of all joint density function $\gamma$ over $\mathcal{T} \times \mathcal{T}$ such that,

$$
\sum_{i=1}^N \gamma(y, y_i) = p_{S_N}(y), \quad \int_{\mathcal{T}} \gamma(y, y') dy = \frac{1}{N} \mathbb{I}(y' = y_i \text{ for some } i \in [n]),
$$

which also implies that $\gamma(y, y') = 0$ as long as $y'$ is not in $\{y^{(1)}, \ldots, y^{(N)}\}$.

$\square$

## C.3 PROOF OF PROPOSITION 1

**Assumption 1.** *We assume that when the solution number $N_1 > N_2$, the packing distance solving Equation (5) is strictly decreasing, i.e., $\delta_{\mathcal{T}}^*(N_1) < \delta_{\mathcal{T}}^*(N_2)$.*

*Proof.* Under this assumption, Let $N_1$ is the maximal packing number of $\delta_{\mathcal{T}}^*(N_1)$. In such a case, the solution set $\mathcal{Y}_{N_1}^*$ is also a $\delta_{\mathcal{T}}^*(N_1)$-covering. Since when it is not a $\delta_{\mathcal{T}}^*(N_1)$-covering, then there exist a solution $y' \in \mathcal{T}, y \neq y_i, i \in [N_1]$ such that $\rho(y^{(i)}, y^{(j)}) > \delta_{\mathcal{T}}^*(N_1), i \neq j$, then $\mathcal{Y}_N' = \{y'\} \cup \mathcal{Y}_{N_1}$ is a $(N_1 + 1)$ packing of $\mathcal{Y}_{N+1}$, which is a contradiction. $\square$

## C.4 UNIFORMITY INDUCED: PREFERENCES TO OBJECTIVES

The key is to show when the function $h$ is a constant mapping. We provide two special cases,

1. The entire preference space $\Omega$ is $\mathcal{S}_1^+$ or $\mathcal{S}_2^+$ . The objective function $f$ is ZDT2 or DTLZ2.

2. The entire preference space $\Omega$ is $\Delta_2$. and the objective function $f$ is DTLZ1.

We prove the first case as an example, and the second one can be proved similarly.

*Proof.* For each preference $\lambda = (\lambda_1, \ldots, \lambda_m), \lambda \in \mathcal{S}_+^{m-1}$, the corresponding solution $y = h(\cdot)$ can be expressed in the form of $(k'(\lambda) \cdot \lambda_1, \ldots, k'(\lambda) \cdot \lambda_m)$, since the function $h(\cdot)$ is an "exact" mapping function. Since $(k'(\lambda) \cdot \lambda_1, \ldots, k'(\lambda) \cdot \lambda_m)$ lies on the Pareto front $k \cdot \mathcal{S}_+^{m-1}$, we have,

$$
\begin{cases}
\sum_{i=1}^m \lambda_i^2 = 1, \\
(k'(\lambda))^2 \sum \lambda_i^2 = k^2.
\end{cases}
\tag{22}
$$

Equation (22) directly leads that $h(\cdot) = k \times (\cdot), \forall \lambda \in \mathcal{S}_+^{m-1}$. Thus, $h(\cdot)$ maps an asymptotically uniform distribution up to a constant, which also is an asymptotically uniform distribution. $\square$

## C.5 PROOF OF LEMMA 1

*Proof.* We demonstrate the uniqueness of the solution $y^*$ by utilizing Lemma 2. According to Lemma 2, the optimal solution of an aggregation function is either Pareto optimal or weakly Pareto optimal. As we have assumed the absence of weakly Pareto optimal solutions, it follows that $y^*$ is the unique solution of Equation (4).

We can furthermore, $y^*$ is a Pareto optimal solution by Lemma 3. ☐

## C.6 PROOF OF THEOREM 1

Blank et al. (2020) generate uniformly distributed preferences $\Lambda_N$, by solving the following optimization problem,

$$\max_{(\Lambda_N \subset \Omega)} \min_{(i,j \in [N], i \neq j)} \rho\left(\lambda^{(i)}, \lambda^{(j)}\right) \tag{23}$$

where $\Omega$ is a compact connected set of $\mathbb{R}^m$. The distance function $\rho(\cdot, \cdot)$ is adopted as the $l_2$ norm in this paper.

To prove Theorem 1, we first need to introduce Lemma 4. According to Borodachov et al. (2007), when $\Omega$ is a rectifiable set [4], we have the following asymptotic uniformity of $\Lambda_N$ when $\Lambda_N$ solve Problem (23),

**Lemma 4.** *For any fixed Borel subset $\mathcal{B} \subseteq \Omega$, one has, when $N \to \infty$,*

$$\mathbb{P}\left(\tilde{\Lambda}_N \in \mathcal{B}\right) = \frac{\texttt{Card}(\Lambda_N \cap \mathcal{B})}{\texttt{Card}(\Lambda_N)} \to \frac{\mathcal{H}_{m-1}(\mathcal{B})}{\mathcal{H}_{m-1}(\Omega)} = \mathbb{P}\left(\tilde{\Lambda} \in \mathcal{B}\right). \tag{24}$$

We use $\texttt{Card}(\cdot)$ to represent the cardinality of a set. Lemma 4 directly follows from the proof of (Borodachov et al., 2007, Theorem 2.2). Let $\tilde{\Lambda}_N$ be a random variable sampled from the category distribution of the set $\Lambda_N$, where each category has a probability of $\frac{1}{N}$. $\tilde{\Lambda}$ is a random variable sampled from the uniform distribution on $\Omega$, denoted as $\text{Unif}(\Omega)$. See Appendix A.1 for more discussions about Hausdorff measure $\mathcal{H}_{m-1}(\cdot)$.

Lemma 4 is equivalent to say that $\tilde{\Lambda}_N \xrightarrow{\text{d}} \text{Unif}(\Omega)$. To prove Theorem 1, according to the continuous mapping theorem (Durrett, 2019, Theorem 3.2.10), $\widetilde{\mathcal{Y}}_N = h \circ \tilde{\Lambda}_N \xrightarrow{\text{d}} h \circ \text{Unif}(\Omega)$.

---

[4] Any compact and connected set with a finite Hausdorff dimension is a rectifiable set. As we have assumed $\Omega$ is compact and connected, then $\Omega$ is rectifiable

