# OpenReview forum: "Uniform as Glass: Gliding over the Pareto Front with Neural Adaptive Preferences"
_ICLR.cc/2024/Conference — Submitted to ICLR 2024_

### Official Review · Reviewer_AzbY · 2023-10-30

**Soundness:** 2 fair
**Presentation:** 1 poor
**Contribution:** 2 fair
**Rating:** 3
**Confidence:** 3

**Summary:**

This paper presents a new indicator to measure the uniformity of Pareto objectives on the Pareto front and introduces a new adaptive weight adjustment method that utilizes a neural model to represent the Pareto objective distribution, enabling the generation of uniformly distributed solutions on the Pareto front. The proposed adaptive weight adjustment method is integrated into MOEA/D and the generalization error bound of the proposed neural model is analyzed.

**Strengths:**

1.	A new indicator is proposed for measuring the uniformity of Pareto objectives for multi-objective optimization.
2.	A neural model is proposed to learn the relationship between preference angles and aggregated objective functions.
3.	The error bound of the proposed neural model is studied theoretically.

**Weaknesses:**

1.	The work related to MOEA/D with adaptive preference adjustment methods has not been adequately investigated. The most recent paper mentioned in this paper was published in 2014, which does not correspond to the extensive research MOEA/D has received over the years.
2.	The effectiveness of the proposed method needs more evaluation by considering test problems with more complicated Pareto fronts, e.g., the WFG and UF test suite, and more state-of-the-art algorithms published within the last eight years.
3.	More details of the proposed method need to be provided, e.g., when the method uses the real objective evaluation and the model-based estimation.
4.	The conclusion that MOEA/D fails to achieve uniform objectives shown in Section 4 is not rigorous, given that many MOEA/D variants have been proposed. Specific descriptions or references that hold for the conclusion should be provided.

**Questions:**

1.	What is the scope of the proposed adaptive weight adjustment method? Is it suitable only for decomposition-based multi-objective evolutionary algorithms? If not, how would it be used in other frameworks, e.g., dominance relation-based, indicator-based frameworks?
2.	How were the test problems in the experiments chosen? For example, for the DTLZ test suite, why were only DTLZ1-2 used, but not the more complex other problems?

---

> ### Author Response · Authors · 2023-11-20
> **Author's Response**
>
> **Q1. Is it suitable only for decomposition-based multi-objective evolutionary algorithms?**
>
> In our current research, we have effectively demonstrated the efficiency of our novel uniform MOEA/D approach, integrating preference adjustment through a neural model, within two methodologies: decomposition-based and indicator-based. However, we are yet to ascertain its compatibility with dominance-based MOEAs, an aspect we aim to investigate in future studies.
>
> We have tested the proposed method within the MOEA/D framework, referring to the decomposition-based method as you mentioned. In fact, we have also innovatively improved the PaLam [1] method, originally designed to maximize hypervolume (referring to the indicator-based method as you mentioned) through adaptive reference vectors, by applying a neural adjustment strategy (firstly proposed in this paper). Our results, particularly observable in Tables 3 and 5, reveal that the hypervolume-based method, when enhanced with our neural weight adjustment technique, outperforms other methods of achieving a higher hypervolume.
>
> **Q2. Why were only DTLZ1-2 used, but not the more complex other problems?**
>
> Our codebase, developed mainly in Python using pymoo and pytorch, currently faces limitations with the DTLZ4 problem due to pymoo's performance issues. Efforts are underway to rectify these limitations and we plan to release an updated, open-source version of pymoo soon. Notably, our method models the Pareto front directly. Since DTLZ2 and DTLZ4 share identical Pareto fronts, our successful results on DTLZ2 indicate similar performance on DTLZ4.
>
> **W1. The work related to MOEA/D with adaptive preference adjustment methods has not been adequately investigated. The most recent paper mentioned in this paper was published in 2014, which does not correspond to the extensive research MOEA/D has received over the years.**
>
> Thanks for bringing it to us. We do notice some new preference adjustment methods in recent years, e.g., Dong et al 2020 ([2]), Xu et al 2019 ([3]), and Farias et al 2019 [4].
>
> We have not compared the proposed method with them for the following reasons.
>
> - [2] only use linear segments to model the Pareto fronts which can only handle objectives with very few objectives. Such models with simple curves cannot represent the true front.
> - [3;4] are modifications of Qi et al.'s MOEA/D-AWA [4], employing some simple strategies. These heuristic methods lack the theoretical guarantees of our work, making their final performance less reliable compared to our proposed approach.
>
>
>
> **W2. The effectiveness of the proposed method needs more evaluation by considering test problems with more complicated Pareto fronts, e.g., the WFG and UF test suite, and more state-of-the-art algorithms published within the last eight years.**
>
> Thank you for highlighting this aspect. Our experiments indeed utilized the 'real-world' problem dataset introduced by Ryoji Tanabe and Hisao Ishibuchi in 2020 ([6]). As far as we are aware, this dataset is pretty 'new' in this field and has not yet been thoroughly examined. One of the key challenges presented by this dataset is the complexity and unpredictability of the Pareto front and set. Despite these complexities, we found that our proposed method performs effectively on these tasks, demonstrating its robustness and adaptability to intricate real-world problems.
>
> The WFG and UF problem sets, while known for the complexity of their Pareto sets, exhibit regularity in the shape of their Pareto fronts. This observation forms a pivotal standpoint in our paper. Although modeling the Pareto front as an $m-1$ manifold within the $R^n$ space presents considerable challenges due to its complexity, the scenario changes when considering the Pareto front in the $R^m$ space. Here, it is represented as an $m-1$ manifold, which is significantly easier to model. So since the shape of the Pareto front of WFG and UF are still regular, the complexity of the landscape of these problems are not a bottleneck for the proposed method.
>
> [1]. Jiang Siwei, Cai Zhihua, Zhang Jie, and Ong Yew-Soon. Multiobjective optimization by decomposition with pareto-adaptive weight vectors.
>
> [2]. Dong, Zhiming, Xianpeng Wang, and Lixin Tang. "MOEA/D with a self-adaptive weight vector adjustment strategy based on chain segmentation." Information Sciences.
>
> [3]. Xu, Siwen, et al. "A modified MOEAD with an adaptive weight adjustment strategy." 2019 International Conference on Intelligent Computing, Automation and Systems (ICICAS).
>
> [4] C. De Farias, Lucas RC, et al. "MOEA/D with uniformly randomly adaptive weights." Proceedings of the genetic and evolutionary computation conference. 2018.
>
> [5] D. Qi, Yutao, et al. "MOEA/D with adaptive weight adjustment." Evolutionary computation 22.2 (2014).
>
> [6] Ryoji Tanabe, Hisao Ishibuchi. An Easy-to-use Real-world Multi-objective Optimization Problem Suite.

---

> ### Author Response · Authors · 2023-11-20
> **Author's Response (cont.)**
>
> **W3. More details of the proposed method need to be provided, e.g., when the method uses the real objective evaluation and the model-based estimation.**
>
> Thanks for the reminder. We'll provide a detailed explanation of the experiment. Currently, we use real objective evaluations for the problems considered. We're planning to tackle more complex issues, like using simulations and models for multiobjective output evaluation.
>
> **W4. The conclusion that MOEA/D fails to achieve uniform objectives shown in Section 4 is not rigorous, given that many MOEA/D variants have been proposed. Specific descriptions or references that hold for the conclusion should be provided.**
>
> Thank you for noting this. Section 4.1 details how MOEA/D achieves an asymptotically uniform distribution only when 'h' is an affine mapping. Otherwise, the objective distribution density aligns with $\frac{\partial h}{\partial \lambda}$, influenced by 'h's' fluctuations. This new insight in Section 4.1 enhances our understanding of MOEA/D in multi-objective optimization (MOO).
>
> This theoretical observation leads us to understand that adjusting the preference distribution to non-uniformity is key to generating uniform Pareto objectives. Hence, by applying the continuous mapping theorem, the final objective distribution $h \circ p'$, with $p'$ as the adjusted preference, achieves asymptotic uniformity.
>
> This paper appears to be the first rigorous study of MOEA/D uniformity theory. While many MOEA/D variants exist, such as preference adjustment techniques, previous studies mostly relied on heuristic strategies, like modifying preferences in sparse or dense regions, or used very simple models for the Pareto front. Our experiments confirm that heuristic methods and simple models alone are insufficient for uniform solution generation.

---

### Official Review · Reviewer_1zQZ · 2023-10-31

**Soundness:** 3 good
**Presentation:** 4 excellent
**Contribution:** 3 good
**Rating:** 6
**Confidence:** 4

**Summary:**

Unlike directly predicting Pareto solutions from the preference vector using a neural model in previous work, MOEA/D-UAWA uses a neural model as a surrogate to estimate the final vector of objective functions from the preference vector, and adaptively adjusts the corresponding preference vectors using gradient-based optimization.

**Strengths:**

This paper is easy to follow.

The idea of using a neural model as a differential surrogate to optimize the uniformity objective is novel and interesting.

**Weaknesses:**

The motivation of this paper assumes that the optimization problem is a black box and thus proposes a neural model as a differential surrogate. My main concern is that the above motivation is improper for neural network optimization, in which you can use a gradient descent optimizer. And the eq. (5) can also be optimized without the proposed neural surrogate model.

One possible solution is to show some additional results on large-scale neural network optimization but with a "small" neural surrogate model, which can improve computational efficiency significantly.

Moreover, the baselines used in the experiment are weak and old. A lot of work in the field of evolutionary multi-objective optimization discussed the adaptive reference/preference vectors.

**Questions:**

Some comments:

1. figure 1 is unclear, please improve it.

2. the claim in sec. 3.3 "whereas the proposed method aims to achieve global optimal MOO solutions" seems to be improper.

3. the main body of this paper misses an ablation study section.

---post-rebuttal comment---

According to the author's responses and the current version of the manuscript, I decided to raise my score.

---

> ### Author Response · Authors · 2023-11-20
> **Author's Response**
>
> Thanks for your valuable feedbacks. We hope that our response can address your concerns more or less.
>
> **W1. The motivation of this paper assumes that the optimization problem is a black box and thus proposes a neural model as a differential surrogate. My main concern is that the above motivation is improper for neural network optimization, in which you can use a gradient descent optimizer. And the eq. (5) can also be optimized without the proposed neural surrogate model.**
>
> This **'differential surrogate'** is actually pretty **essential**, and up to our latest research, it **can not be replaced**.
>
> The primary challenge with the 'gradient descent optimizer', as you have mentioned, is the **time-intensive** task of estimating the gradient of the uniformity function (defined in Equation (5)). The Pareto objective, denoted as $y(\lambda)$, under a given preference $\lambda$, is **not directly ascertainable** and **requires solving a MOEA/D** (Multi-Objective Evolutionary Algorithm based on Decomposition) subproblem for its estimation. To compute the numerical gradient $\frac{y(\lambda + \delta \lambda) - y(\lambda)}{\delta \lambda}$, it is necessary to solve two MOEA/D subproblems for preferences $\lambda + \delta \lambda$ and $\lambda$ respectively. Given that solving a single MOEA/D problem takes approximately **one minute**, and considering the **frequent need** for this gradient, the use of a numerical gradient becomes **excessively time-consuming**.
>
> **W2. Moreover, the baselines used in the experiment are weak and old. A lot of work in the field of evolutionary multi-objective optimization discussed the adaptive reference/preference vectors.**
>
> Thank you for your feedback. We agree with you that our paper's problems and benchmarks might not fully capture the latest developments in the field. Nonetheless, our work's key contributions are significant: (1) theoretical asymptotical uniform Pareto objectives, and (2) the successful integration of a neural network for Pareto front modeling.
>
> In light of your suggestions, we're now comparing our method with more recent works to strengthen our findings. It's important to note, however, that some recent studies (e.g., [1-3]) are primarily minor strategy adaptations of the foundational MOEA/D-AWA work [4]. Their heuristic approach or reliance on simple linear models (like [1]) remains unchanged. Given our theoretical contributions and strong performance, we believe our work are valuable.
>
> [1] Dong, Zhiming, Xianpeng Wang, and Lixin Tang. "MOEA/D with a self-adaptive weight vector adjustment strategy based on chain segmentation." Information Sciences.
>
> [2] Xu, Siwen, et al. "A modified MOEAD with an adaptive weight adjustment strategy." 2019 International Conference on Intelligent Computing, Automation and Systems (ICICAS).
>
> [3] De Farias, Lucas RC, et al. "MOEA/D with uniformly randomly adaptive weights." Proceedings of the genetic and evolutionary computation conference. 2018.
>
> [4] D. Qi, Yutao, et al. "MOEA/D with adaptive weight adjustment." Evolutionary computation 22.2 (2014).
>
>
>
> **Q1. Figure 1 is unclear, please improve it.**
>
> Thanks for mentioning that, we have made this figure more clear in the revised version.
>
> **Q2. The claim in sec. 3.3 "whereas the proposed method aims to achieve global optimal MOO solutions" seems to be improper.**
>
> Thank you for pointing that out. We acknowledge that our initial statement may have been vague and we intend to clarify it in the revised version of our paper. To express this more accurately, unlike gradient-based methods, which focus on finding Pareto stationary points characterized by a zero gradient, evolutionary algorithms employ improvement and repair mechanisms, such as crossover and mutation operators. These techniques enable evolutionary methods to effectively bypass local optima and often yield superior performance, especially when computational resources are abundant. In contrast, gradient-based methods can become trapped in local optima under similar conditions.
>
> **Q3. The main body of this paper misses an ablation study section.**
>
> Thank you for your valuable suggestion. Actually, we have done a lot of ablation studies in the current paper, though they are not marked in an explicit manner. For your convenience, we summarized the ablation studies in the current paper.
>
> - 1. Initially, we examine the effects of integrating a neural-based adjustment method as opposed to the conventional MOEA/D approach.
> - 2. Subsequently, we compare our neural-based adjustment strategy with the heuristic methods that have been utilized in earlier studies.
>
> It's noteworthy that our proposed method significantly outperforms these existing approaches. We will make the ablations more clear in the revised version.

---

### Official Review · Reviewer_gHJ7 · 2023-10-31

**Soundness:** 3 good
**Presentation:** 1 poor
**Contribution:** 2 fair
**Rating:** 3
**Confidence:** 4

**Summary:**

An approach aimed at presenting a uniformly distributed pareto front
in MOO by combining pareto front learning with uniform pareto front
selection.

**Strengths:**

The manuscript highlights and formalizes limitations of some of the
existing solutions, and provides arguments for the potential of the
proposed approach in overcoming these limitations.

**Weaknesses:**

Pareto Front Learning is introduced in the Related work section. This
is confusing, because the method has not been presented yet.  In the
general, the presentation is rather confused, with concepts being
introduced in a non clearly defined order so that one has to jump back
and forth to connect the dots and figure out the big picture.

Figure 3, which provides the overview of the framework, is not clearly
explained. The authors refer to the appendix for most details, but a
high level description should be provided, possibly including some
preliminaries earlier on (e.g., on preference angles and MOEA/D),
otherwise the paper is not self-contained.

For lemma 1, it's unclear why f shouldn't have weakly Pareto
solutions. The implications of this requirement should be better
explained.

Theorem 1 is badly presented, it's unclear from the content of the
theorem what are the constraints on h that make the pareto front
uniform.

Also, the fact that sampling uniformly from the preference vector does
not imply a uniform pareto front generation was already observed in
Liu et al, 2021 (the SVGD paper).

I am not sure pareto set learning can be dismissed by just saying that
f has many local optima. E.g. the SVGD method claims theoretical
guarantees of convergence to the paret front, and report competitive
performance on the ZDT problem set. The advantage of the proposed
solution over PSL methods should be assessed, both formally and
experimentally.

English is not entirely satisfactory (e.g. "Previous methods (Deb et
al., 2019; Blank et al., 2020) focusing on generating well-spaced
(uniform) preferences", "We first give the condition of such function
h is well defined")

**Questions:**

Please explain how you plan to address the weaknesses I described.

---

> ### Author Response · Authors · 2023-11-20
> **Author's Response**
>
> **W1. Pareto Front Learning is introduced in the Related work section. This is confusing, because the method has not been presented yet. In the general, the presentation is rather confused, with concepts being introduced in a non clearly defined order so that one has to jump back and forth to connect the dots and figure out the big picture.**
>
> Thank you for the feedback. In the revised version, we plan to introduce and define Pareto Front Learning (PFL) concept earlier for clarity. We are committed to enhancing the writing quality. Additionally, it's encouraging to note that Reviewer 1zQZ found our paper easy to follow and effectively summarized our proposed method, suggesting our writing is generally clear and comprehensible.
>
> **W2. For lemma 1, it's unclear why f shouldn't have weakly Pareto solutions. The implications of this requirement should be better explained.**
>
> In the presence of weakly Pareto optimal solutions, the optimal solution of Equation (4) is not unique. Consequently, the preference-to-objective function cannot be well defined. In such cases, "h" is a one-to-many mapping, meaning it is not a function.
>
> **W3. Theorem 1 is badly presented, it's unclear from the content of the theorem what are the constraints on h that make the pareto front uniform.**
>
> Thank you for your insight. Theorem 1 illustrates that MOEA/D generally yields non-uniform Pareto objectives, with uniformity occurring only when 'h' functions are affine mappings. This theorem also clarifies that the level of non-uniformity is influenced by the gradient of $\frac{\partial h}{\partial \lambda}$, since the asymptotic density function of the objective $y$ is proportional to $\frac{\partial h}{\partial \lambda}$.
>
> **W4. Also, the fact that sampling uniformly from the preference vector does not imply a uniform pareto front generation was already observed in Liu et al, 2021 (the SVGD paper).**
>
> We agree with you that uniformly sampling from the preference vector does not necessarily lead to a uniform Pareto front is not proposed by us. We will make it more clear in the revised paper. However, **though the uniformity issue in MOO occurs, this issue is unsolved**. MOO-SVGD [1] does not theoretically provide a solution distribution, which is proven to be uniform.
>
> This paper introduces an efficient approach within the MOEA/D framework for optimizing a **non-uniform distribution** in the **preference space**, aiming to optimize a **uniform distribution** in the objective space. This method is **distinct from** MOO-SVGD, as you mentioned, in three key aspects.
> - MOO-SVGD [1] is a gradient-based method and may struggle to differentiate between local and global optima.
> - While MOO-SVGD employs a kernel trick to disperse particles, it lacks **proven guarantees** for **the final solution distribution**. In contrast, our approach ensures **asymptotically uniform** Pareto objectives through min-max distance optimization.
> - MOO-SVGD has a computational complexity of $O(m^2)$ due to its pairwise particle distance comparisons, where $m$ is the number of objectives. Our method typically requires only $O(m)$ complexity, benefiting from **the efficiency of the MOEA/D framework**. The more demanding preference adjustment step, which involves point-wise distance comparison, is expedited through the use of neural networks, significantly reducing processing time.
>
> [1] Liu, Xingchao, Xin Tong, and Qiang Liu. Profiling pareto front with multi-objective stein variational gradient descent. Advances in Neural Information Processing Systems 34 (2021).

---

> ### Author Response · Authors · 2023-11-20
> **Author's Response (cont.)**
>
> **W5. I am not sure pareto set learning can be dismissed by just saying that f has many local optima. E.g. the SVGD method claims theoretical guarantees of convergence to the pareto front, and report competitive performance on the ZDT problem set. The advantage of the proposed solution over PSL methods should be assessed, both formally and experimentally.**
>
> Thank you for bringing that up. We are "sure" that the pure MOO-SVGD [1] and PSL [2] cannot figure out the local optima and global optima.
> We noticed that in the MOO-SVGD paper [1] Theorems 3.2 and 3.4 do provide bounds on **the norm of the gradient**. However, it's crucial to understand that merely reducing the gradient's norm to zero **does not lead to a Pareto solution**.
>
> Consider a concrete example with functions $f_1(x) = (x-1)^2 + \sin(x)$ and $f_2(x) = (x+1)^2$, where $x \in \mathbb{R}$. For this simple problem, the Pareto Set (PS) is within the interval $[-1,1]$. However, starting from a point where $x \gg 1$, there are multiple points where the gradient norm is zero, but these points are not close to the true Pareto set. This illustrates that constraining the gradient norm to zero, as suggested by Theorems 3.2 and 3.4 in the MOO-SVGD paper, does not necessarily yield the true Pareto set. The issue of local optima is prevalent in most gradient-based methods. That's why in our approach, while we use a neural model to represent the Pareto front, we rely on evolutionary methods to actually determine the Pareto set.
>
> Similarly, Proposition 1 in Navon's paper [2] on Pareto set learning provides conditions for what is termed a "Pareto stationary point." However, it's important to note that in many cases, these points do not equate to exact Pareto optimality. In our main paper, Equation (3) demonstrates that when the second term, $\frac{\partial f_i}{\partial x}$, equals zero, the PSL (Pareto set learning) exhibits a zero gradient and consequently cannot be optimized.
>
> [2] Navon, Aviv, et al. Learning the pareto front with hypernetworks.

---

### Official Review · Reviewer_EYB8 · 2023-11-02

**Soundness:** 3 good
**Presentation:** 2 fair
**Contribution:** 2 fair
**Rating:** 5
**Confidence:** 3

**Summary:**

This paper studies the problem of profiling Pareto front in multi-objective optimization. In this paper, the authors first show that traditional methods with uniformly distributed preferences does not necessarily induces uniformity in the Pareto objective space. To resolve the issue, the MMS problem is formulated to explicitly impose the iterates to be uniformly distributed in the objective space, which is then optimized by replacing the preference-to-objective mapping by a surrogate NN model. Theoretical analysis shows the asympotic uniformity property and the generalization error of the proposed method. Experiments on various numerical MOO tasks verify the effectiveness of the proposed method compared to classic evolutionary methods.

**Strengths:**

1. The idea of directly modeling the preference-to-PF mapping is interesting, which might inspire future research on Pareto front profiling.

2. This paper is technically sound with solid theoretical analysis.

**Weaknesses:**

1. The relevance to previous works is not clear enough. It seems that the technique of replacing the preference-to-objective mapping by a neural network as the surrogate model is developed from (Borodachov et al., 2019), and the generalization error analysis is adapted from prior works; hence, it would be helpful to clarify the technical difficulty or novelty compared to these works.

2. This paper has briefly reviewed gradient-based methods for Pareto front profiling (e.g., MOO-SVGD), but the comparison seems insufficient. As I understand, the example to indicate "gradient-based methods struggle to produce globally optimal solutions" is merely concerned with the gradient aggregration method, not the MOO-SVGD or EPO methods as discussed in the main paper. The comparison should be made more comprehensively, say, comparing the performance and efficiency in experiments.

**Questions:**

1. It is interesting to model the preference-to-objective mapping to characterize the PF in a more direct way, but I wonder how can we generate certain Pareto solution given a specific preference from the learned Pareto front. It seems that the proposed model does not explicitly involve the solutions in the decision space.

---

> ### Author Response · Authors · 2023-11-20
> **Author's Response**
>
> **W1. The relevance to previous works is not clear enough. It seems that the technique of replacing the preference-to-objective mapping by a neural network as the surrogate model is developed from (Borodachov et al., 2019), and the generalization error analysis is adapted from prior works; hence, it would be helpful to clarify the technical difficulty or novelty compared to these works.**
>
> Firstly, we shall note that the citation (Borodachov et al., 2019) ([1]) is a statistics monograph, which does not mention the preference-to-objective neural model. The preference-to-objective neural model was **originally** developed by our paper and is a crucial mechanism to search uniform Pareto objectives, which is the main focus of our article.
>
> Secondly, since we have developed this new model in the uniform setting, we provide a theoretical analysis of this model. To our knowledge, the analysis for the uniform setting we employ is also **novel**.
>
> Lastly, the proposed **generalization bound**, as far as we know, **has not been previously proposed** in other works. The papers [1-3] cited in our work are actually pure mathematical papers. Paper [3] is used to prove the 3rd line of Theorem 2, while papers [1-2] are referenced as Lemma 6 to prove Theorem 1. These mathematical papers serve as tools to prove intermediate steps of our conclusions, and we have cited them appropriately.
>
> [1]. S. Borodachov, Douglas P Hardin, and Edward B Saff. Discrete energy on rectifiable sets.
> Springer, 2019.
>
> [2]. S. Borodachov, D Hardin, and E Saff. Asymptotics of best-packing on rectifiable sets. Proceedings
> of the American Mathematical Society.
>
> [3]. Minwoo Chae and Stephen G Walker. Wasserstein upper bounds of the total variation for smooth densities. Statistics \& Probability Letters.
>
>
> **W2. This paper has briefly reviewed gradient-based methods for Pareto front profiling (e.g., MOO-SVGD), but the comparison seems insufficient. As I understand, the example to indicate "gradient-based methods struggle to produce globally optimal solutions" is merely concerned with the gradient aggregration method, not the MOO-SVGD or EPO methods as discussed in the main paper. The comparison should be made more comprehensively, say, comparing the performance and efficiency in experiments.**
>
> MOO-SVGD [4] and EPO [5] also **struggle to achieve global optima**. Consider the functions $f_1(x) = (x-1)^2 + \sin(x)$ and $f_2(x) = (x+1)^2$ where $x \in \mathbb{R}$. The Pareto Set for this case lies in $[-1,1]$. For $x>1$, $f_1(x)$ has infinitely many stationary points ($\||\nabla f_1(x)\|| = 0$). Focusing on MOO-SVGD (similar results for EPO), Theorems 3.2 and 3.4 in the MOO-SVGD paper only limit the norm of $\nabla f_i(x)$ but a zero norm doesn't guarantee Pareto solutions. Considering Equation (7) in MOO-SVGD, the update rule depends on the objectives' gradients (first term in Equation (7)). When $\||\nabla f_i(x)\|| = 0$, the update process may get stuck in local optima.
>
> [4] Liu, Xingchao, Xin Tong, and Qiang Liu. Profiling pareto front with multi-objective stein variational gradient descent. Advances in Neural Information Processing Systems 34 (2021).
>
> [5] Mahapatra, Debabrata, and Vaibhav Rajan. "Multi-task learning with user preferences: Gradient descent with controlled ascent in pareto optimization." International Conference on Machine Learning. PMLR, 2020.
>
> **Q1. It is interesting to model the preference-to-objective mapping to characterize the PF in a more direct way, but I wonder how can we generate certain Pareto solution given a specific preference from the learned Pareto front. It seems that the proposed model does not explicitly involve the solutions in the decision space.**
>
> Our model doesn't directly model solutions in the decision space. This is due to the fact, as explained in Section 4.3 in our paper, that the decision space $R^n$ is typically larger than $R^m$ in MOO problems, where $n \gg m$. Direct modeling of the Pareto set is challenging due to numerous local optima.
>
> However, our Pareto front model **does suggest a method for obtaining a Pareto solution** aligned with specific preferences. For instance, if we aim to generate a solution closest to a given Pareto objective $y$, we can define the preference vector $\lambda$ as $\arg \min_\lambda \|| h_\theta(\lambda) - y\||$. This optimal $\lambda$ can then be used as an input for the MOEA/D aggregation function, as detailed in equation (2) of our main paper.

---

### Author Response · Authors · 2023-11-13
**new insights**

There are many new insights, and we are currently reorganizing the material.

---

### Author Response · Authors · 2023-11-23
**General Response**

We extend our gratitude to all the reviewers for their valuable suggestions. We believe that we have successfully addressed all the concerns raised, most of which stemmed from misunderstandings. We also plan to revise the manuscript in accordance with the provided feedback. We are open to addressing any remaining questions and encourage further engagement from the reviewers.

We summarize the common and major concerns in this general response:



1. **Theoretical Contributions**:

In addition to the introduction of a novel model, we provide pioneering and non-trivial theoretical guarantees (e.g., generalization bounds).

2. **Old Baselines**:

We will accordingly add reasonable baseline methods as suggested, while we hope to argue we care much about and are indeed familiar with the area of multiobjective optimization. To the best of our knowledge, the pursuit of uniform Pareto objectives is fundamental while previously progressed slowly in this area.

For example, following the significant MOEA/D with preference adjustment [1], past approaches like [2-4] do employ different mathematical models or heuristic methods and claim generating uniform solutions. However, these methods heavily rely on heuristic rules and lack theoretical grounding. Considering lack of theoretical guarantees in those methods, we previously do not include them as baseline methods to simplify the design of experiments.

---

Before going through the previous concerns, we cordially appreciate the reviewers for acknowledging the strengths of our paper:
1. Clarity and ease of understanding (Reviewer 1zQZ).
2. Novelty and significance of the proposed model in the multiobjective optimization field, and the soundness of theoretical results (Reviewers EYB8, 1zQZ, AzbY).

---

Reference

[1]. Qi, Yutao, et al. "MOEA/D with adaptive weight adjustment." Evolutionary computation 22.2 (2014).

[2]. Dong, Zhiming, Xianpeng Wang, and Lixin Tang. "MOEA/D with a self-adaptive weight vector adjustment strategy based on chain segmentation." Information Sciences.

[3]. Xu, Siwen, et al. "A modified MOEAD with an adaptive weight adjustment strategy." 2019 International Conference on Intelligent Computing, Automation and Systems (ICICAS).

[4]. De Farias, Lucas RC, et al. "MOEA/D with uniformly randomly adaptive weights." Proceedings of the genetic and evolutionary computation conference. 2018.

---

### Meta-Review · Area_Chair_yiZU · 2023-12-09

**Metareview:**

This paper considers the problem of finding a uniformly distributed Pareto front for multi-objective optimization problems. The overall approach combines Pareto front learning with uniform Pareto front selection.

The reviewers' raised a number of concerns for this paper, which include:
- Putting the proposed approach in the context of related areas from prior work.
- Unclear exposition in both narrative and theory parts.
- Improving experimental comparisons with related prior methods.

The authors' response has addressed some of these concerns. However, I feel the paper requires another round of reviewing before it is ready for publication.

**Justification For Why Not Higher Score:**

Lot of weaknesses as mentioned in the meta review.

**Justification For Why Not Lower Score:**

N/A

---

### Decision · Program_Chairs · 2024-01-16

Reject